# Cooperative cobinding of synthetic and natural ligands to the nuclear receptor PPARγ

Jinsai Shang[1], Richard Brust[1], Sarah A Mosure[1,2,3], Jared Bass[1], Paola Munoz-Tello[1], Hua Lin[4], Travis S Hughes[5,6], Miru Tang[7], Qingfeng Ge[7], Theodore M Kamenekca[4], Douglas J Kojetin[1,4]*

[1]Department of Integrative Structural and Computational Biology, The Scripps Research Institute, Jupiter, United States; [2]Summer Undergraduate Research Fellows (SURF) program, The Scripps Research Institute, Jupiter, United States; [3]Skaggs Graduate School of Chemical and Biological Sciences, The Scripps Research Institute, Jupiter, United States; [4]Department of Molecular Medicine, The Scripps Research Institute, Jupiter, United States; [5]Center for Biomolecular Structure and Dynamics, The University of Montana, Missoula, United States; [6]Department of Biomedical and Pharmaceutical Sciences, The University of Montana, Missoula, United States; [7]Department of Chemistry and Biochemistry, Southern Illinois University, Carbondale, United States

**Abstract** Crystal structures of peroxisome proliferator-activated receptor gamma (PPARγ) have revealed overlapping binding modes for synthetic and natural/endogenous ligands, indicating competition for the orthosteric pocket. Here we show that cobinding of a synthetic ligand to the orthosteric pocket can push natural and endogenous PPARγ ligands (fatty acids) out of the orthosteric pocket towards an alternate ligand-binding site near the functionally important omega (Ω)-loop. X-ray crystallography, NMR spectroscopy, all-atom molecular dynamics simulations, and mutagenesis coupled to quantitative biochemical functional and cellular assays reveal that synthetic ligand and fatty acid cobinding can form a 'ligand link' to the Ω-loop and synergistically affect the structure and function of PPARγ. These findings contribute to a growing body of evidence indicating ligand binding to nuclear receptors can be more complex than the classical one-for-one orthosteric exchange of a natural or endogenous ligand with a synthetic ligand.
DOI: https://doi.org/10.7554/eLife.43320.001

*For correspondence:
dkojetin@scripps.edu

Competing interests: The authors declare that no competing interests exist.

## Introduction

Thiazolidinediones (TZDs), also called glitazones, are synthetic agonists of the peroxisome proliferator-activated receptor gamma (PPARγ) nuclear receptor with potent anti-diabetic efficacy in patients with type 2 diabetes (*Soccio et al., 2014*). TZDs were initially discovered in phenotypic screens as compounds with anti-diabetic efficacy (*Fujita et al., 1983*), and later studies identified PPARγ as the molecular target (*Lehmann et al., 1995*). After an increase in clinical use of TZD drugs such as rosiglitazone (*Avandia*) in the early 2000s, the use of TZDs in diabetic patients has dropped significantly due to adverse side effects, including bone loss and water retention. These observations have spawned efforts to determine the molecular mechanisms by which TZDs and other synthetic PPARγ ligands exert beneficial antidiabetic effects and adverse effects.

Crystal structures of the PPARγ ligand-binding domain (LBD) have been critical in determining the molecular mechanism of action of PPARγ-binding ligands. Overall, the structure of the PPARγ LBD is

described as a three-layer helical sandwich fold composed of 12 α-helices containing a dynamic or flexible internal ligand-binding pocket. The PPARγ ligand-binding pocket (*Figure 1A*) is among the largest in the nuclear receptor superfamily (*Gallastegui et al., 2015*), ranging from 1200 to 1400 Å$^3$ in crystal structures with three branches described as 'T' or 'Y' shaped (*Kroker and Bruning, 2015*). Branch I is located proximal to helix 12, which forms a critical part of the activation function-2 (AF-2) coregulator binding surface; branch II is located near the β-sheet surface, close to the predicted ligand entry/exit site where helix 3 meets helix 6 and the helix 11/12 loop; and branch III is also located near the β-sheet surface but deeper in the pocket near helix 5 (*Edman et al., 2015*; *Aci-Sèche et al., 2011*; *Genest et al., 2008*). One of the first crystal structures of PPARγ was solved bound to the synthetic TZD ligand rosiglitazone (*Nolte et al., 1998*), which revealed critical details on the molecular contacts formed between a ligand and the binding pocket of the receptor. Subsequent crystal structures of PPARγ bound to dietary medium-chain fatty acids (MCFAs) and long-chain unsaturated fatty acids (UFAs), the latter of which are endogenous PPARγ ligands (*Kim et al., 2011*), revealed overlapping binding modes with synthetic ligands that occupy all three branches of the orthosteric pocket (*Kroker and Bruning, 2015*; *Itoh et al., 2008*; *Li et al., 2008a*; *Waku et al., 2009a*; *Waku et al., 2009b*; *Malapaka et al., 2012*). These crystal structures and others, combined with biochemical ligand displacement assays (*Kliewer et al., 1997*; *Ziouzenkova et al., 2003*; *Xu et al., 1999*; *Nagy et al., 1998*), revealed that synthetic and natural ligands compete for binding to the same orthosteric pocket.

However, recent studies have shown that synthetic PPARγ ligands that were originally designed to bind to the orthosteric pocket can also bind to an alternate site (*Hughes et al., 2016*; *Li et al., 2008b*; *Hughes et al., 2014*; *Brust et al., 2017*; *Jang et al., 2017*; *Bae et al., 2016*). The alternate site partially overlaps with one of the arms of the T/Y-shaped orthosteric pocket near the β-sheet surface (branch II), but it uniquely occupies space in a solvent exposed pocket formed by the flexible Ω-loop that precedes helix 3 (*Figure 1B*). Structural studies have linked ligand occupancy of branch I (i.e., the helix 12 subpocket) within the orthosteric pocket to robust stabilization of the AF-2 surface,

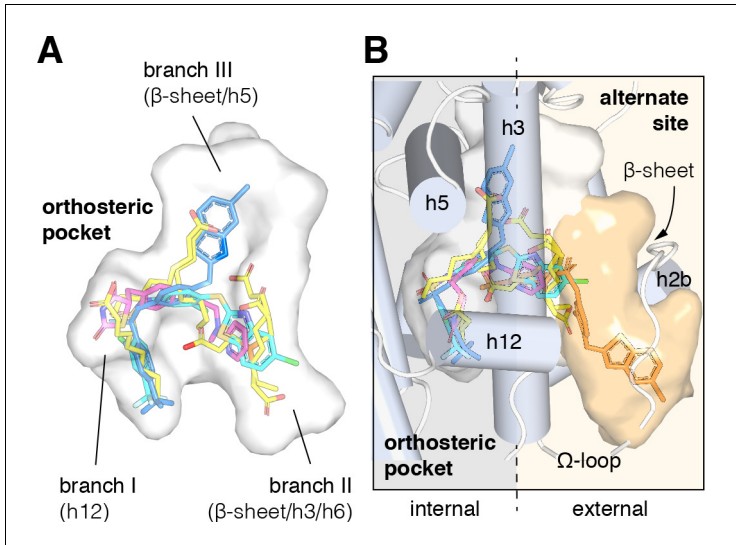

**Figure 1.** Structural locations of PPARγ orthosteric pocket and alternate site. (**A**) The T/Y-shaped orthosteric pocket can accommodate one or more natural ligands such as nonanoic acid (C9; yellow sticks; PDB: 4EM9) and synthetic ligands such as rosiglitazone (pink sticks; PDB: 4EMA) or T2384 (light and dark blue sticks representing different crystallized binding modes; PDB: 3K8S, chain A and B, respectively). (**B**) The orthosteric pocket (white pocket surface) is completely enclosed within the alpha helical sandwich fold of the ligand-binding domain (LBD). Ligands such as T2384 (orange sticks; PDB: 3K8S, chain B) can also bind to a solvent accessible alternate site (orange pocket surface) distinct from the orthosteric pocket, structurally defined as the region between helix 3 (h3) and the flexible Ω-loop (dotted line separating the region internal to the LBD, grey background; and external to the LBD, light orange background).
DOI: https://doi.org/10.7554/eLife.43320.002

increased 'classical' transcriptional activation, and side effects associated with TZDs in part driven by hydrogen bonds formed between the TZD head group and residues within branch I (*Choi et al., 2010*). In contrast, ligands that either do not form hydrogen bonds in the helix 12/branch I subpocket, or bind to a combination of branch II and the alternate site, can stabilize the Ω-loop region and a nearby loop. Stabilization of this region near the Ω-loop can inhibit phosphorylation of a serine residue (S273 in PPARγ isoform one or S245 in isoform 2) by the Cdk5 kinase and is associated with anti-diabetic efficacy (*Jang et al., 2017*; *Bae et al., 2016*; *Choi et al., 2011*; *Choi et al., 2014*).

We were initially interested in the structural changes that allows the TZD edaglitazone (*Dietz et al., 2012*; *Fürnsinn et al., 1999*) to fit within the branch I/helix 12 subpocket. Edaglitazone contains a bulky benzo[b]thiophene group adjacent to the TZD head group, in contrast to other TZD-bound PPARγ crystal structures, which have smaller moieties (e.g., phenyl) adjacent to the TZD head group. We solved the crystal structure of the PPARγ LBD bound to edaglitazone at the orthosteric pocket, which unexpectedly revealed a bacterial MCFA cobound to the alternate site. MCFAs are not only bacterial lipids; they are also dietary mammalian natural PPARγ ligands obtained from foods such as oils and dairy products (*Rial et al., 2016*). MCFAs display high serum concentrations (μM–mM) that activate PPARγ transcription; regulate the expression of PPARγ target genes; and influence cellular differentiation, adipogenesis, and insulin sensitization in mice (*Malapaka et al., 2012*; *Liberato et al., 2012*; *Nagao and Yanagita, 2010*). The cobound MCFA provides a 'ligand link' to the flexible Ω-loop, and crystal structures of PPARγ bound to covalent orthosteric antagonists revealed that synthetic ligands extensions within the orthosteric pocket can push fatty acid cobinding modes towards the Ω-loop. Inspired by these findings, we solved several crystal structures showing that endogenous UFAs can be pushed from their orthosteric binding modes to an alternate site cobinding mode when a synthetic TZD ligand binds to the orthosteric pocket. Using NMR spectroscopy, molecular dynamics simulations, isothermal titration calorimetry (ITC), and mutagenesis coupled to functional assays, we show that cobinding of a synthetic ligand and fatty acid affects the conformation of the AF-2 surface, synergistically enhances coactivator interaction, and may affect PPARγ transcription in cells.

## Results

### Edaglitazone is a TZD PPARγ agonist

Edaglitazone is similar in chemical structure to a commonly used PPARγ agonist, rosiglitazone (*Figure 2A*), both of which possess the prototypical TZD head group. Edaglitazone has a bulkier 5-methyl-2-phenyloxazole tail group compared to the N-methylpyridin-2-amine tail group in rosiglitazone. Furthermore, there is a substitution of the phenyl moiety adjacent to the TZD head group in rosiglitazone for a bulkier benzo[b]thiophene moiety in edaglitazone. Edaglitazone and rosiglitazone display similar affinities for PPARγ, 141 nM and 93 nM respectively, in a fluorescent tracer ligand displacement assay (*Figure 2B*). The compounds also show similar potencies in a TR-FRET biochemical assay on increasing the interaction of a peptide derived from the TRAP220 coactivator (*Figure 2C*; 132 nM and 186 nM respectively) and decreasing the interaction with a peptide derived from the NCoR corepressor (*Figure 2D*; 171 nM and 432 nM respectively); as well as in a cell-based transcriptional reporter assay (*Figure 2E*; 5.4 nM and 3.2 nM respectively).

### Crystal structure of PPARγ cobound to edaglitazone and a MCFA

We obtained crystals of the PPARγ ligand-binding domain (LBD) by co-crystallizing protein incubated with edaglitazone. Crystals formed in the I4 space group, a different space group compared to the >150 crystal structures of PPARγ deposited in the PDB. The structure was solved at 2.1 Å (*Table 1*) as a homodimer with two molecules in the asymmetric unit (chains A and B) with the expected alpha helical sandwich fold. In the absence of a coregulator peptide, helix 12 typically adopts two distinct conformations in chains A and B in previously solved crystal structures—an 'active' conformation, where helix 12 is docked into the AF-2 surface; and an atypical conformation via contacts to another symmetry related molecule within the unit cell. However, in our edaglitazone-bound structure, both chains adopt an active helix 12 conformation (*Figure 3A*) with nearly identical backbone conformations (Cα r.m.s.d. 0.186 Å).

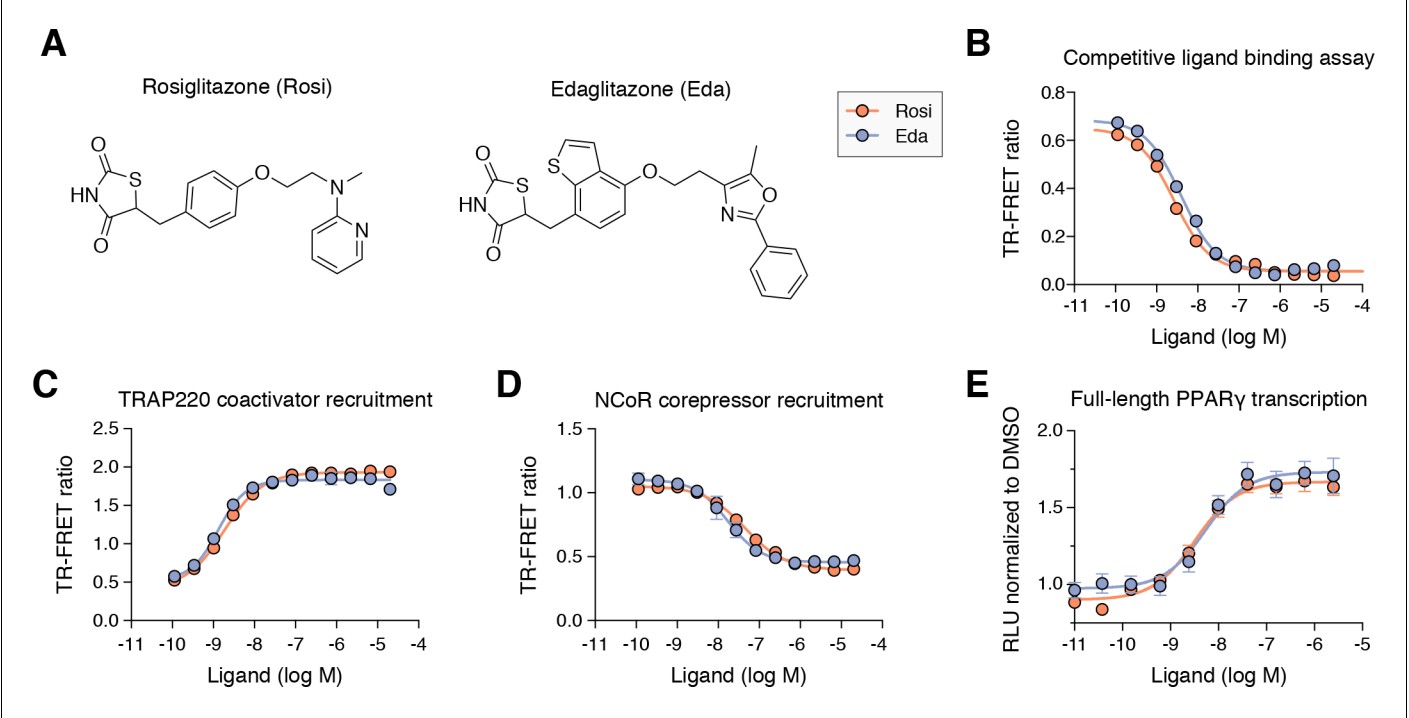

**Figure 2.** Edaglitazone is a thiazolidinedione (TZD) PPARγ agonist related to rosiglitazone. (**A**) Chemical structures of edaglitazone and rosiglitazone. (**B**) TR-FRET ligand displacement assay. Data plotted as the average and S.D. of three experimental replicates. (**C,D**) TR-FRET coregulator interaction assay data in the presence of peptides derived from (**C**) TRAP220 coactivator and (**D**) NCoR corepressor. Data plotted as the average and S.D. of three experimental replicates. (**E**) Cell-based luciferase reporter assay measuring full-length PPARγ transcription in HEK293T cells; RLU = relative luciferase units. Data plotted as the average and S.E.M. of four experimental replicates.

DOI: https://doi.org/10.7554/eLife.43320.003

Strong electron density was observed for three ligands in both chain A and B molecules (*Figure 3B*; *Figure 3—figure supplement 1*): one edaglitazone bound to the canonical or orthosteric pocket; a second edaglitazone to a surface pocket created through an interaction between two symmetry-related protein molecules (*Figure 3—figure supplement 2*); and a MCFA consistent with the size of a nonanoic acid (C9) molecule bound to a solvent accessible region near helix 3, the β-sheet surface of the orthosteric pocket, and the flexible Ω-loop region. Notably, a previous study showed that PPARγ expressed in *E. coli* can retain MCFAs from the bacteria, predominately C9, which function as natural PPARγ agonists (*Malapaka et al., 2012*; *Liberato et al., 2012*). The combined binding of the C9 and the second edaglitazone may have contributed to the stability of the PPARγ Ω-loop, which is often characterized with very poor or absent electron density in most published crystal structures.

When bound to the orthosteric pocket, rosiglitazone (*Figure 3C*) and edaglitazone (*Figure 3D*) make conserved hydrogen bonds via the TZD head group to the side chains of S289 (helix 3), H323 (helix 5), H449 (helix 10/11), and Y473 (helix 12). The TZD head group of rosiglitazone also makes a hydrogen bond to Q286 (helix 3), which is not present in the edaglitazone-bound structure due to a 2.8 Å shift of the Q286 side chain from the rosiglitazone-bound position (5.7 Å away from the TZD 2-carbonyl position). This occurs due to a rearrangement of the N-terminal region of helix 3 in the edaglitazone-bound structure, which affects the conformation of Q286 more than S289 (*Figure 3E*). Additional conformational changes in the edaglitazone-bound structure move helix 6 and the loop connecting helix 6–7 away from the ligand-binding pocket, and the movement of helix 3 also causes a concomitant movement of helix 12. Finally, F363 and M364 are found in the loop connecting helix 6–7 with secondary structure elements classified as a hydrogen bonded turn and bend, respectively according to DSSP analysis (*Kabsch and Sander, 1983*). These residues undergo a shift of 6.0 Å (Phe363) and 2.6 Å (M364) away from the ligand-binding pocket to accommodate the benzo[b]

**Table 1.** X-ray crystallography data collection and refinement statistics.

| | Edagtliazone (+ C9) |
|---|---|
| Data collection | |
| Space group | I4 |
| Cell dimensions | |
| a, b, c (Å) | 128.74, 128.74, 93.67 |
| α, β, γ (°) | 90, 90, 90 |
| Resolution | 33.36–2.1 (2.18–2.1) |
| $R_{pim}$ | 0.049 (0.299) |
| I / σ(I) | 9.87 (2.83) |
| CC1/2 in highest shell | 0.798 |
| Completeness (%) | 99.31 (99.69) |
| Redundancy | 2.0 (2.0) |
| | |
| Refinement | |
| Resolution (Å) | 2.10 |
| No. of unique reflections | 44385 |
| $R_{work}/R_{free}$ (%) | 18.2/21.6 |
| No. of atoms | |
| Protein | 4354 |
| Water | 473 |
| B-factors | |
| Protein | 27.30 |
| Ligand | 29.14 |
| Water | 33.73 |
| Root mean square deviations | |
| Bond lengths (Å) | 0.008 |
| Bond angles (°) | 1.18 |
| Ramachandran favored (%) | 96.28 |
| Ramachandran outliers (%) | 0.93 |
| PDB accession code | 5UGM |

*Values in parentheses indicate highest resolution shell.
DOI: https://doi.org/10.7554/eLife.43320.007

thiophene in the edaglitazone-bound structure, along with several other subtler changes (<1 Å) that accommodate 5-methyl-2-phenyloxazole tail group near the β-sheet (*Figure 3F*).

## MCFAs cobound in other PPARγ crystal structures bound to synthetic ligands

A previous study reported a crystal structure of PPARγ LBD that was obtained using protein purified from *E. coli*, which revealed the presence of bound endogenous bacterial MCFAs (*Liberato et al., 2012*). Specifically, three molecules of C9 were observed in the orthosteric pocket (*Figure 4A*) overlapping in space where edaglitazone is bound in our structure (*Figure 4B*). In our edaglitazone-bound crystal structure, there is clear density for a bacterial MCFA near helix 3, the β-sheet surface of the orthosteric pocket, and the flexible Ω-loop region. This region is referred to as an 'alternate' or 'allosteric' site, because other studies have shown that synthetic ligands that have crystallized in the orthosteric pocket can also bind to this alternate site. Alternate site binding can occur via a second molar equivalent of ligand, for example one bound to the orthosteric pocket and a second to the alternate site (*Hughes et al., 2016*; *Li et al., 2008b*; *Hughes et al., 2014*); when the orthosteric

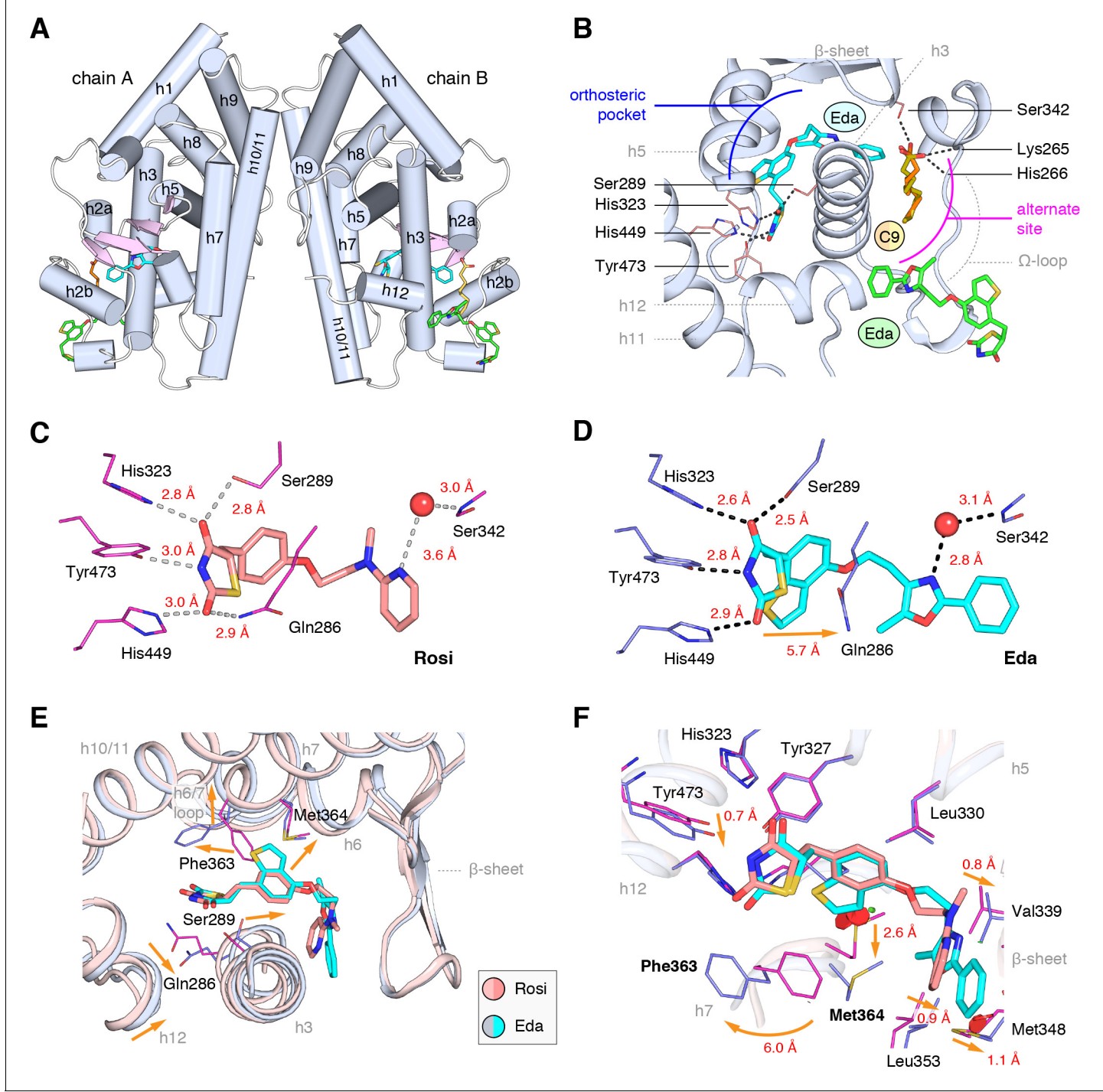

**Figure 3.** Co-crystal structure of edaglitazone-bound PPARγ LBD reveals a cobound bacterial MCFA. (**A**) Overall structure (helices, light blue; strands, pink) with two bound edaglitazone (EDA) ligands, one to the canonical orthosteric pocket (blue) and another to a surface pocket (green), and a C9 ligand bound to an alternate site (yellow and orange in chain A and B, respectively). (**B**) Molecular interactions between PPARγ and the EDA and C9 ligands. Dotted lines denote hydrogen bonds. (**C,D**) Comparison of the hydrogen bond interactions between PPARγ and (**C**) rosiglitazone (PDB: 2PRG) and (**D**) edaglitazone. Dotted lines denote hydrogen bonds. (**E,F**) Conformational changes relative to rosiglitazone-bound PPARγ (PDB: 2PRG) that allow the orthosteric pocket to adapt to binding the bulkier benzo[b]thiophene moiety in edaglitazone. The changes (orange arrows) include (**E**) shifting of the backbone of helix 3, helix 6, helix 6/7 loop, and helix 12; and (**F**) large movements of the Phe363 and M364 side chains adjacent to the benzo[b] thiophene moiety. *Also see Figure 3—figure supplements 1 and 2.*

DOI: https://doi.org/10.7554/eLife.43320.004

*Figure 3 continued*

The following figure supplements are available for figure 3:

**Figure supplement 1.** Omit map (2F$_O$–F$_C$, contoured at 1 σ) of the EDA and C9 ligands in the crystal structure.
DOI: https://doi.org/10.7554/eLife.43320.005

**Figure supplement 2.** The TZD head group of the second edaglitazone molecule docks into the PPARγ AF-2 surface.
DOI: https://doi.org/10.7554/eLife.43320.006

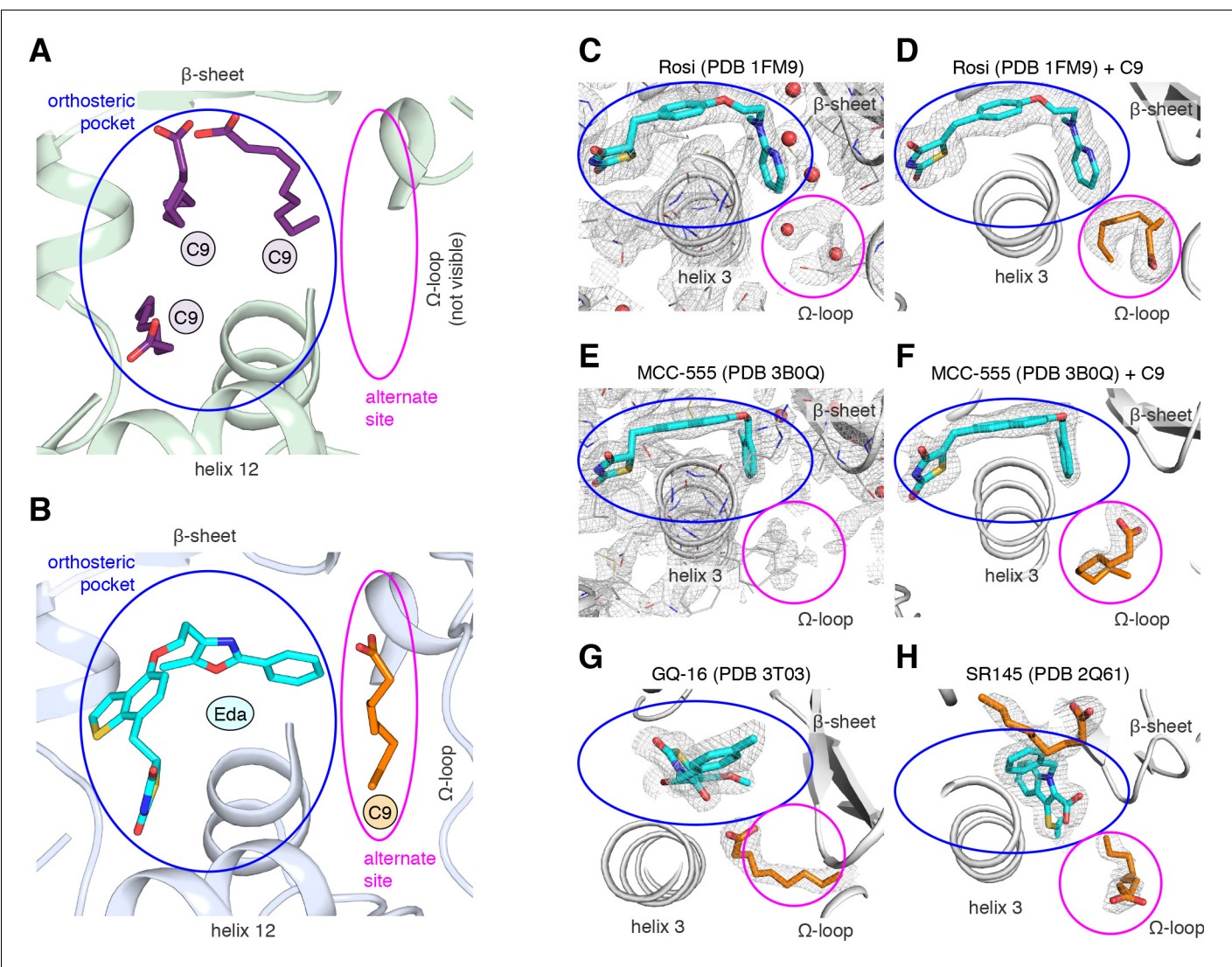

**Figure 4.** MCFA electron density in previously solved synthetic ligand-bound PPARγ LBD crystal structures. (**A,B**) Comparison of a crystal structure of PPARγ LBD with (**A**) three molecule of C9 bound to the orthosteric pocket (PDB: 4EM9) and (**B**) C9 bound to the alternate site with edaglitazone bound to the orthosteric pocket. (**C–F**) Electron density maps for previously determined TZD-bound (blue) crystal structures shown with (**C,E**) original deposited structural models and (**D,F**) C9 (orange) modeled and refined into density present in the alternate site. (**G,H**) C9 (orange) modeled and refined into previously determined crystal structures of PPARγ bound to (**G**) GQ-16 (blue) and (**H**) SR145 (blue). Omit maps (2F$_O$–F$_C$) contoured at 1 σ in all but the C9 modeled into the GQ-16 structure, which was contoured at 0.5 σ. The orthosteric pocket and alternate site are circled in blue and magenta, respectively.
DOI: https://doi.org/10.7554/eLife.43320.008

pocket is blocked by a covalent antagonist inhibitor that physically blocks the orthosteric pocket (*Hughes et al., 2016*; *Hughes et al., 2014*; *Jang et al., 2017*; *Bae et al., 2016*); or one equivalent through an alternate binding mode (*Jang et al., 2017*; *Bae et al., 2016*).

We wondered whether there could be density present for a bacterial MCFA in previously published crystal structures of PPARγ bound to synthetic ligands, which may not have been modeled during structure determination because it could have been unexpected at the time. Analysis of the electron density maps deposited to the PDB for two relatively high resolution (~2 Å) crystal structures bound to rosiglitazone (PDB: 1FM6) and MCC-555 (PDB: 3B0Q) revealed extra unidentified densities suggesting there could be a MCFA bound in a similar manner to what we observed in the edaglitazone structure. In a crystal structure of RXRα-PPARγ LBD heterodimer where PPARγ was bound to rosiglitazone (PDB: 1FM6) solved to 2.1 Å (*Gampe et al., 2000*), two water molecules were modeled next to rosiglitazone where the C9 is bound in our edaglitazone structure (*Figure 4C*). We found that replacement of the water molecules with a C9 better describes the density in this region (*Figure 4D*). In a crystal structure of PPARγ bound to MCC-555 (PDB: 3B0Q) solved to 2.1 Å, we also found unmodeled density (*Figure 4E*) consistent with a MCFA present in the same region (*Figure 4F*). Additionally, we found density consistent with a MCFA in crystal structures of PPARγ bound to GQ-16 (*Figure 4G*) and SR145 (*Figure 4H*), which had been solved to 2.1 Å and 2.2 Å, respectively (*Bruning et al., 2007*; *Amato et al., 2012*). These observations suggest that cobound bacterial MCFAs may be present in other synthetic ligand-bound PPARγ crystal structures. Importantly, MCFAs are also dietary natural ligands present in oils and dairy products consumed by mammals; thus, the fortuitous discovery that bacterial MCFAs can cobind with synthetic orthosteric ligands has structural and functional relevance to PPARγ.

## NMR validates cobinding of C9 with a TZD bound to the orthosteric pocket

To determine the degree to which bacterial MCFAs are bound to PPARγ, we performed differential 2D [$^1$H,$^{15}$N]-TROSY HSQC NMR analysis of $^{15}$N-labeled PPARγ LBD subjected to a delipidation protocol and 'native' protein that was not subjected to delipidation (*Figure 5—figure supplement 1*), which revealed subtle chemical shift and peak intensity/line broadening changes. Notably, titration of C9 (*Figure 5—figure supplement 2*) into native PPARγ caused additional selective NMR chemical shift changes, indicating that native protein is likely only partially bound to (i.e., is not saturated with) bacterial MCFAs. We have observed that, compared to delipidated protein, native NMR chemical shift changes are batch dependent. We also found that native PPARγ NMR spectra can appear nearly identical to delipidated PPARγ NMR spectra, indicating that the amount of bacterial lipid that may be bound and retained during purification can vary from substoichiometric to almost none. These changes can result from how the protein purification is performed, such as the length of chromatography or dialysis steps, or variability of fatty acid content depending on the bacterial growth media used, as the use of minimal vs. rich media can change exogenous fatty acid content within bacteria (*Tao et al., 1999*).

We also performed differential NMR analysis of rosiglitazone-bound forms of native and delipidated PPARγ to determine if there could be cobinding of a synthetic TZD to native protein bound to a substoichiometric amount of a bacterial MCFA. Although the NMR spectra overlay very closely (*Figure 5—figure supplement 3*), subtle NMR chemical shift perturbations were observed for a select number of peaks (*Figure 5A*), including residues in our crystal structure near the C9 bound to the alternate site on helix 3 (V277, Q283, and G284), as well as residues within the AF-2 surface on helix 5 (L317 and K319) and helix 12 (L465, I472, and K474). Furthermore, several NMR peaks with weak intensities in the native form showed narrower, more intense peaks in the delipidated form (*Figure 5A*; 'new' label). In our previous NMR analysis of rosiglitazone-bound PPARγ (*Hughes et al., 2012*), we were unable to assign these peaks due to their weak and/or absent peaks in 3D NMR data, likely due to the substoichiometric cobinding of bacterial MCFAs.

We next performed differential NMR analysis to determine if exogenously added MCFAs, which are dietary components of the mammalian diet, can cobind with synthetic ligands to PPARγ. We titrated C9 into rosiglitazone-bound native PPARγ (*Figure 5B*) and delipidated PPARγ (*Figure 5C*) and again performed differential NMR analysis. The binding affinity of C9 (*Figure 5—figure supplement 4*) is ~50,000 fold weaker than rosiglitazone; thus, to eliminate competition for the orthosteric pocket, we titrated one and three molar equivalents of C9 into PPARγ pretreated with two molar

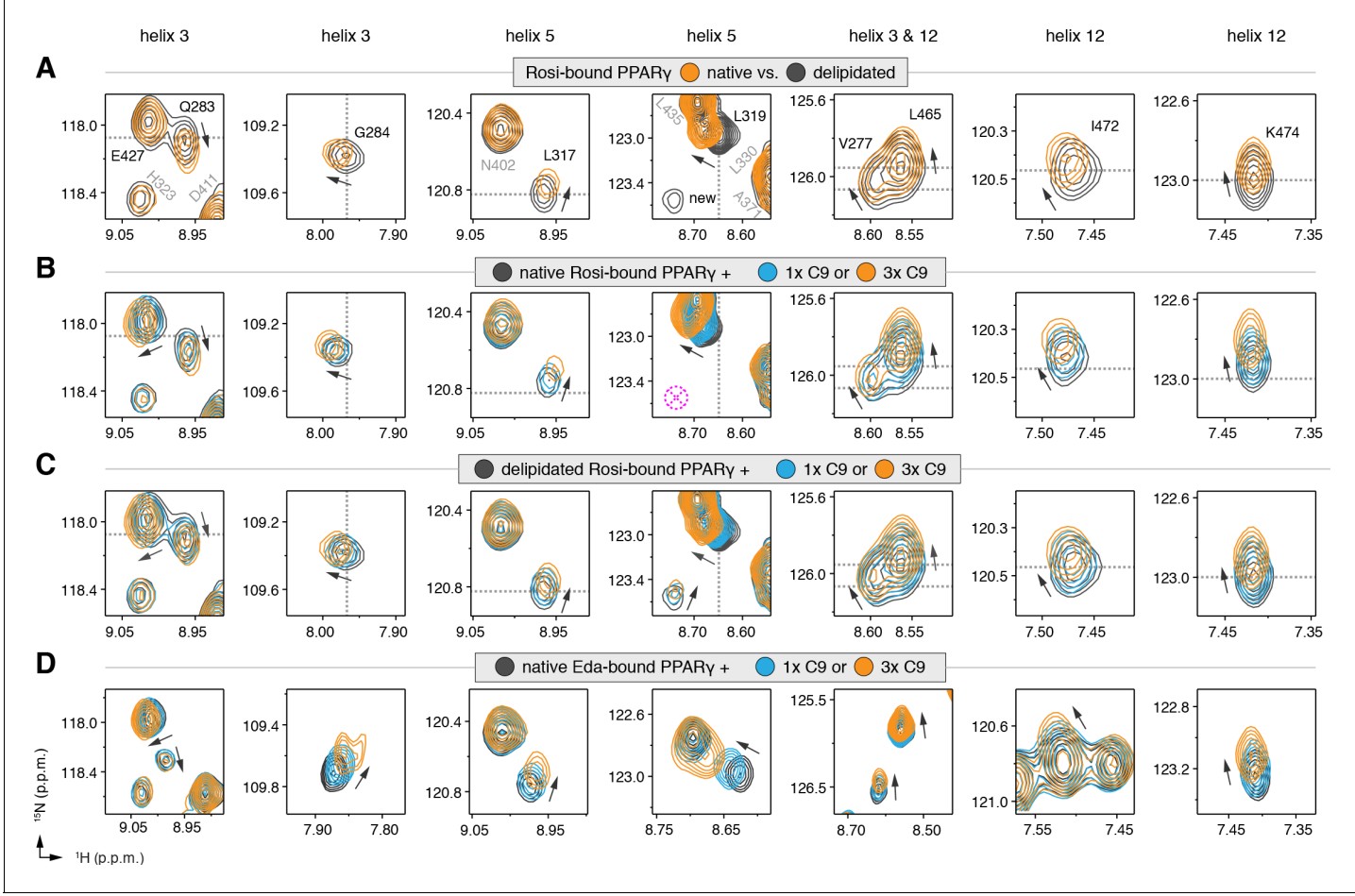

**Figure 5.** NMR confirms cobinding of exogenously added C9 and synthetic TZDs. Differential 2D [$^1$H,$^{15}$N]-TROSY-HSQC NMR data comparing (**A**) native vs. delipidated PPARγ LBD bound to rosiglitazone (two equiv.); (**B**) C9 added into native PPARγ LBD bound to rosiglitazone (two equiv.); (**C**) C9 added into delipidated PPARγ LBD bound to rosiglitazone (two equiv.); and (**D**) C9 added into native PPARγ LBD bound to edaglitazone (two equiv.). The dotted lines in panels A–C represent the vertical ($^1$H chemical shift) or horizontal ($^{15}$N chemical shift) position of the peaks in the delipidated form to illustrate how the peaks corresponding to the native, nondelipidated form shift 'on path' to changes caused by C9 binding. Black arrows denote the chemical shift changes; pink dotted circle/cross denotes a missing peak; and dotted lines denote the delipidated apo-PPARγ peak positions to illustrate the 'on path' transitions of native and C9-bound forms. *Also see Figure 5—figure supplement 1–5.*

DOI: https://doi.org/10.7554/eLife.43320.009

The following figure supplements are available for figure 5:

**Figure supplement 1.** Full differential 2D [$^1$H,$^{15}$N]-TROSY-HSQC NMR data comparing $^{15}$N-PPARγ LBD in the native form (orange) vs. delipidated using a refolding procedure (black) or treatment using LIPIDEX 1000 resin (green), the latter two of which are nearly identical and show an increase in the intensities of some peaks vs. native.

DOI: https://doi.org/10.7554/eLife.43320.010

**Figure supplement 2.** Full differential 2D [$^1$H,$^{15}$N]-TROSY-HSQC NMR data comparing C9 (nonanoic acid; 1, 2, 3, and four equiv.; light to dark blue) added into native $^{15}$N-PPARγ LBD (red).

DOI: https://doi.org/10.7554/eLife.43320.011

**Figure supplement 3.** Full differential 2D [$^1$H,$^{15}$N]-TROSY-HSQC NMR data comparing native (red) vs.delipidated (black) PPARγ bound to rosiglitazone (two equiv.).

DOI: https://doi.org/10.7554/eLife.43320.012

**Figure supplement 4.** TR-FRET based PPARγ LBD ligand displacement assay for nonanoic acid (C9; $K_i$ = 47.4 μM), decanoic acid (C10; 19.4 μM), and dodecanoic acid (C12; 4.1 μM).

DOI: https://doi.org/10.7554/eLife.43320.013

**Figure supplement 5.** Differential 2D [$^1$H,$^{15}$N]-TROSY-HSQC NMR data showing that chemical shifts corresponding to residues (G361, L255, and an unassigned peak) perturbed when.

DOI: https://doi.org/10.7554/eLife.43320.014

equivalents of rosiglitazone. This analysis again revealed subtle NMR chemical shift perturbations for essentially the same residues that were affected in the differential analysis of rosiglitazone bound to native or delipidated forms without added C9 (*Figure 5A*). Notably, the direction of the chemical shift change that occurs upon titration of C9 to rosiglitazone-bound native (*Figure 5B*) or delipidated (*Figure 5C*) PPARγ is the same direction of the shift that occurs when comparing rosiglitazone-bound native and delipidated PPARγ without added C9 (*Figure 5A*). In fact, the peak position of rosiglitazone-bound native PPARγ shows a 'head start' in shifting compared to delipidated PPARγ, which further supports that native rosiglitazone-bound PPARγ is bound to a substoichiometric amount of MCFA. We observed a similar shifting of peaks for edaglitazone-bound PPARγ (*Figure 5D*).

If C9 was competing with rosiglitazone bound to the orthosteric pocket, residues near the TZD head group such as H323, which forms a side chain-mediated hydrogen bond with the aromatic TZD head group, would likely shift significantly. Aromatic ring current effects, which have a significant influence on NMR peak positions, would change in this portion of the orthosteric pocket if C9 displaced rosiglitazone. However, only very minor shifting is observed for the H323 NMR peak as compared to the larger shifting that occurs for residues near the bound C9 in our crystal structure. Furthermore, other NMR peaks affected by C9 binding to PPARγ in the absence of rosiglitazone are not affected in the cobinding experiments with rosiglitazone, nor do they move from the rosiglitazone-bound peak position to the C9-bound peak position (*Figure 5—figure supplement 5*). These observations support the cobinding of C9, not orthosteric competition of C9, under our experimental conditions.

## Molecular simulations reveal conformational effects afforded by ligand cobinding

Our NMR results indicate that C9 cobinding to the alternate site when a TZD is bound to the orthosteric pocket can affect the conformation of the AF-2 surface. We performed molecular simulations to determine the potential conformational consequences of the cobinding of edaglitazone and C9. Two microsecond simulations were conducted with edaglitazone bound to the orthosteric pocket in the absence or presence of C9 at the alternate site. For each ligand-bound state, two replicate simulations on the chain A and B crystallized conformations were performed resulting in a total of eight stable simulations (*Figure 6A*) allowing averaging of up to four simulations for each state. In all eight simulations, edaglitazone remained stably bound to the orthosteric pocket (*Figure 6B*). C9 remained stably bound to the alternate site in three out of four simulations with edaglitazone cobound to the orthosteric pocket (*Figure 6C*). The simulation where C9 exchanged out of the alternate site was therefore excluded from the analysis.

The conformation of the Ω-loop has been shown to be important in regulating the conformation of helix 12, a critical structural element in the AF-2 surface (*Waku et al., 2009a*; *Waku et al., 2009b*; *Puhl et al., 2012*; *Waku et al., 2010*). Conformational fluctuations in the Ω-loop when bound to edaglitazone alone, calculated as the coordinate R.M.S.D. relative to the crystallized conformation after minimization and equilibration, range between 4–6 Å (*Figure 6D*). However, when cobound to C9, Ω-loop conformational fluctuations show a broader range of smaller R.M.S.D. values (2–5 Å) with a notable increase in the 2–4 Å range relative to edaglitazone bound alone. Thus, a conformational stabilizing effect is observed for Ω-loop residues that contact the cobound C9. The loop connecting helix 1 and helix 2a, which is proximal to the Ω-loop and contains a residue (S245 in PPARγ1; S273 in PPARγ2) that is phosphorylated by the kinase Cdk5 (*Choi et al., 2010*), also shows reduced conformational fluctuation when cobound to C9 relative to edaglitazone bound alone (*Figure 6E*). NMR analysis reveals that C9 cobinding with rosiglitazone or edaglitazone affects the conformation of residues near S245 (*Figure 6—figure supplement 1*). These observations support studies showing that ligand occupancy of the alternate site can affect the conformation of this surface and inhibit Cdk5-mediated phosphorylation of S245 (*Bae et al., 2016*). Finally, C9 cobinding to the alternate site stabilizes the conformation of helix 12, which shows decreased R.M.S.D. values relative to edaglitazone bound alone (*Figure 6F*). This is consistent with our NMR data (*Figure 5*) showing that cobinding of C9 with rosiglitazone or edaglitazone affects NMR peaks for residues within the helix 12/AF-2 surface.

We analyzed the hydrogen bond network in the molecular simulations to establish the binding determinants of C9 cobinding with edaglitazone. Several hydrogen bonds between the carboxylate

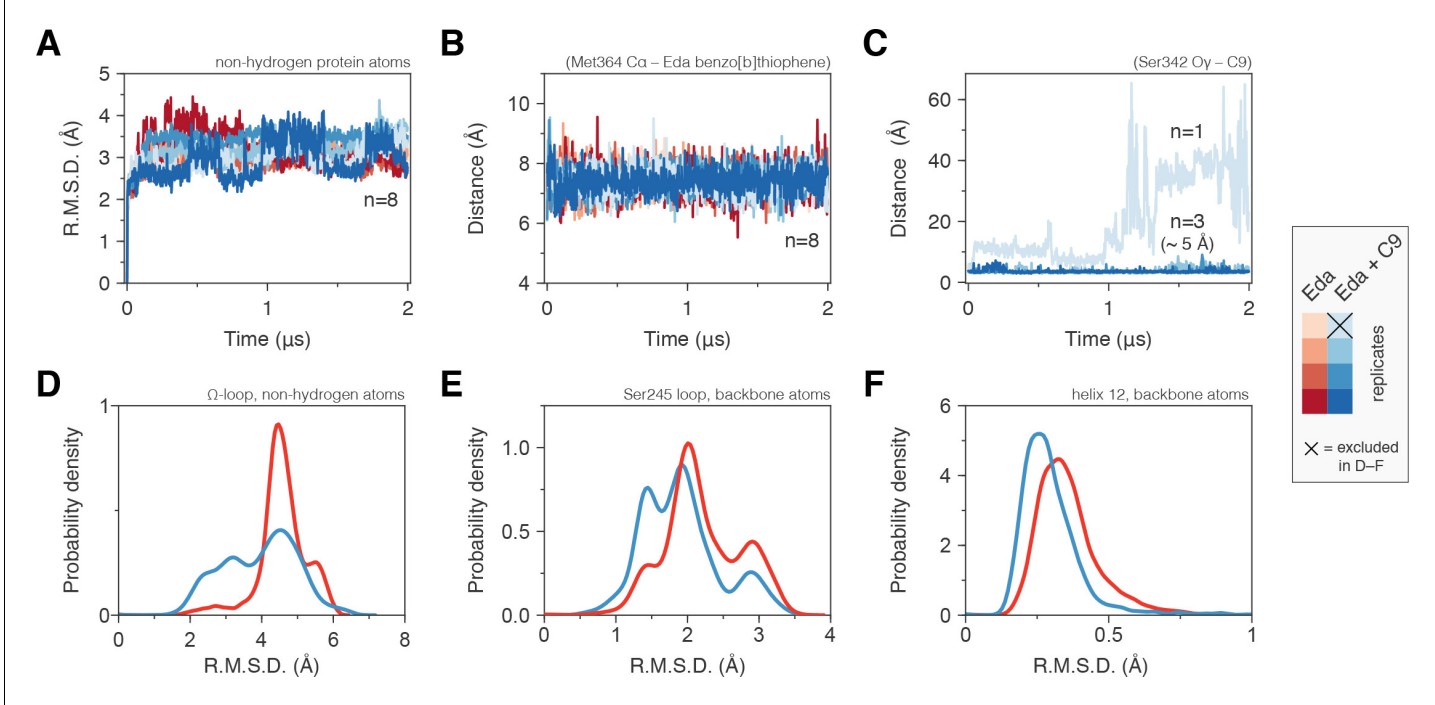

**Figure 6.** Molecular simulations reveal C9 cobinding stabilizes the PPARγ LBD. (**A**) Heavy atom R.M.S.D. to the starting conformation (crystal structure after minimization and equilibration) reveal that all four simulations bound to edaglitazone only to the orthosteric site (red lines) or cobound to C9 at the alternate site (blue lines) are stable during the simulation period. (**B**) Distance between benzo[b]thiophene group in edaglitazone and the M364 Cα atom shows that edaglitazone remains stably bound to the orthosteric pocket in all eight simulations. (**C**) C9 remains stably bound to the alternate site in three of four cobound simulations. In the legend, the × denotes the Eda + C9 (light blue) simulation where C9 unbound during the simulation, which was excluded from the analysis. (**D–F**) Probability density histogram distributions of the R.M.S.D. relative to the starting structure for (**D**) the Ω-loop, (**E**) the loop containing the serine residue phosphorylated by Cdk5, and (**F**) helix 12. *Also see Figure 6—figure supplement 1*.

DOI: https://doi.org/10.7554/eLife.43320.015

The following figure supplement is available for figure 6:

**Figure supplement 1.** Differential 2D [$^1$H,$^{15}$N]-TROSY-HSQC NMR data focusing on residues in the Ser273 loop comparing C9 (1 and 3 equiv.; blue and orange, respectively) added into.

DOI: https://doi.org/10.7554/eLife.43320.016

head group of C9 and various backbone and side chain groups of PPARγ were observed in all three simulations (*Figure 7A*), and among these the hydrogen bonds to the K265 backbone and S342 side chain were also observed in the crystal structure. The hydrogen bond to the H266 backbone observed in the crystal structure was lowly populated in the simulation, and new hydrogen bonds were detected with the R288 side chain. Notably, a complex network of water-bridged hydrogen bonds was observed (*Figure 7B*), including several different types of water-mediated hydrogen bonds linking the oxazole nitrogen atom in edaglitazone to the carboxylate head group of C9 (*Figure 7C*). In addition to these hydrogen bonds, a network of hydrophobic interactions was observed in all three simulations (*Figure 7D*), including the side chains of L255, F264, V277, R280, I281, F287, and the terminal phenyl group of edaglitazone interacts with the saturated carbon side chain of C9 (*Figure 7E*).

## Isolation of MCFA cobinding with covalent orthosteric antagonists

GW9662 (*Figure 8—figure supplement 1*) is an antagonist that covalently binds to C285 located in, and blocks ligand binding to, the orthosteric pocket (*Leesnitzer et al., 2002*). Structural superposition of an available crystal structure of GW9662-bound PPARγ (PDB: 3B0R) with a crystal structure of PPARγ bound to C9 (32) (PDB: 4EMA) revealed that GW9662 overlaps with two out of the three crystallized C9 binding modes in branches I and III (*Figure 8A*). We solved two new crystal structures of PPARγ bound to GW9662 and SR16832 (*Brust et al., 2017*), an analog of GW9662 that we

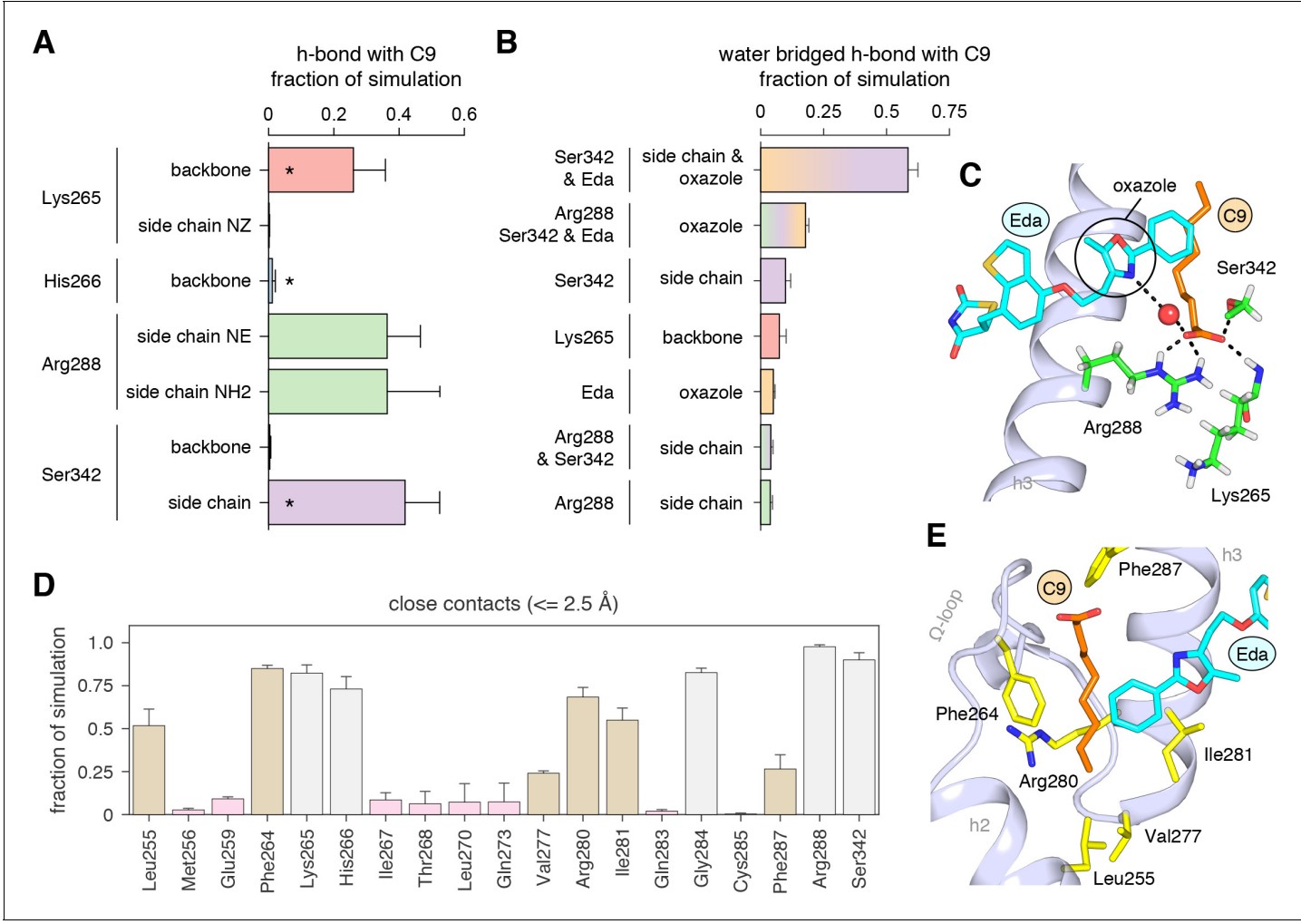

**Figure 7.** Interactions driving cobinding of C9 to the alternate site. (A) Direct and (B) water-bridged hydrogen bonds detected in the three 2 μs molecular simulations where C9 remained stably bound; * denotes hydrogen bonds observed in crystal structure. Data plotted as the average and S.D. of three experimental replicates. (C) Example of the C9-mediated hydrogen bond network from a snapshot of the simulation trajectory. (D) Close atomic contacts populated in the simulation (brown, hydrophobic side chain contacts; grey, other contacts; pink, lower abundance contacts). Data plotted as the average and S.D. of three experimental replicates. (E) Structural example of the hydrophobic contacts (brown in D).
DOI: https://doi.org/10.7554/eLife.43320.017

developed with a ligand extension that weakens synthetic ligand binding to the alternate site (*Brust et al., 2017*), at 2.29 Å and 2.73 Å, respectively (*Table 2*). Similar to our edaglitazone crystal structure, we observed electron density consistent with a bound bacterial MCFA in both structures, which we modeled as C9. In chain A of our GW9662-bound structure, one C9 is present within the branch II region of the orthosteric pocket; in chain B, there are two C9 molecules present, one in branch II and another closer to the cobound position in our edaglitazone structure (*Figure 8B*; *Figure 8—figure supplement 2A*). In the SR16832 structure, C9 is present only in chain A; and the 6-methoxyquinoline group extension in SR16832, relative to GW9662, appears to have blocked the branch II C9 binding mode, pushing the C9 closer to the cobound position in our edaglitazone structure (*Figure 8C*; *Figure 8—figure supplement 2B*).

We performed differential NMR analysis to confirm the C9 binding regions in the crystal structures of PPARγ bound to GW9662 and SR16832. This analysis was limited relative to the analysis we performed above for rosiglitazone and edaglitazone because about half of the expected NMR peaks are missing due to intermediate exchange on the NMR time scale from significant μs–ms time scale dynamics present when bound to covalent antagonists relative to agonists that robustly stabilize the

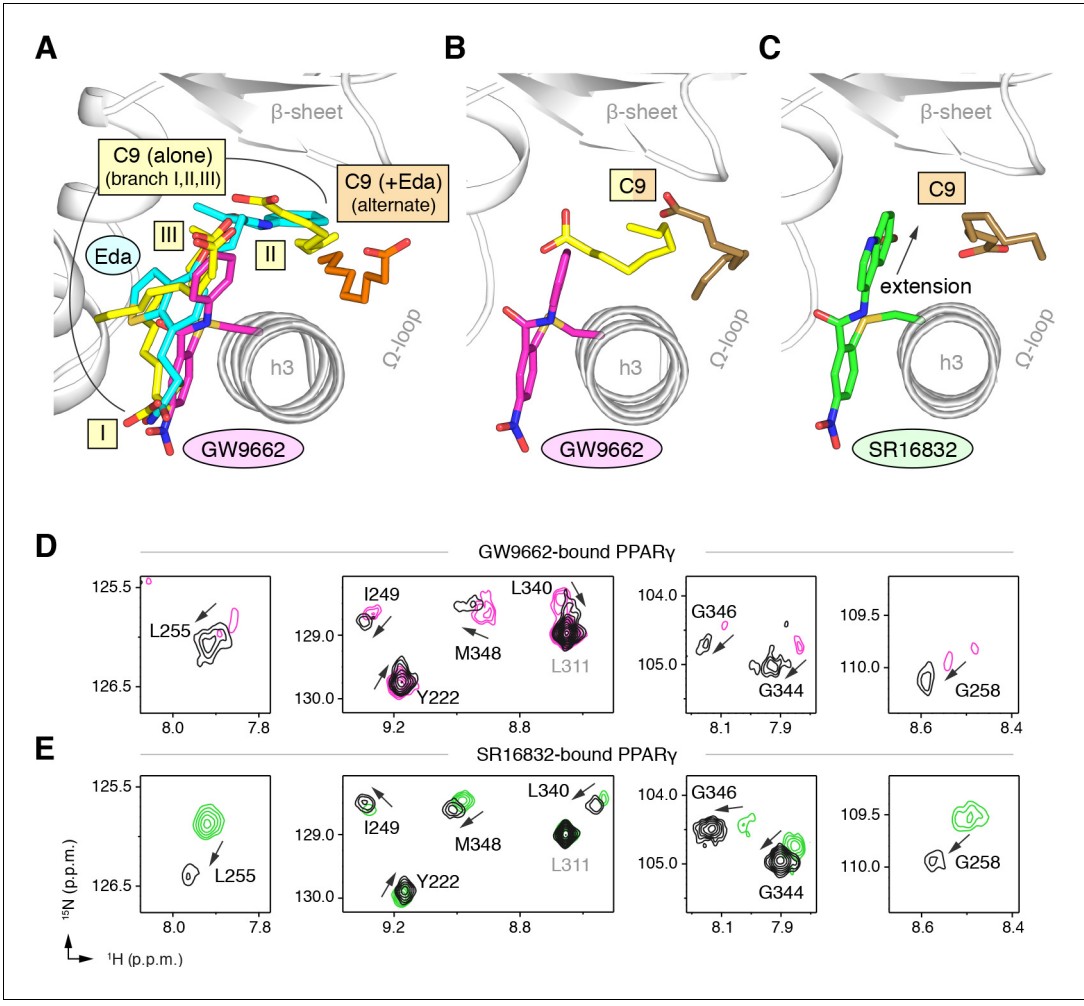

**Figure 8.** Covalent orthosteric antagonists isolate C9 cobinding. (A) The covalent antagonist GW9662 (PDB: 3B0R; magenta) overlaps with the crystallized binding modes of 2 out of 3 C9 molecules bound to the orthosteric pocket (PDB: 4EM9) but not C9 bound to the alternate site. (B,C) Crystal structures of PPARγ LBD bound to covalent antagonists (B) GW9662 and (C) SR16832 cobound to C9. (D,E) Differential 2D [$^1$H,$^{15}$N]-TROSY-HSQC NMR data of PPARγ LBD covalently bound to (D) GW9662 (magenta) or (E) SR16832 (green) confirms cobinding of C9 (black; two equiv.); arrows denote chemical shift changes. *Also see Figure 8—figure supplement 1 and 2.*
DOI: https://doi.org/10.7554/eLife.43320.018

The following figure supplements are available for figure 8:

**Figure supplement 1.** Chemical structures of covalent orthosteric ligands used in this study.
DOI: https://doi.org/10.7554/eLife.43320.019

**Figure supplement 2.** Omit map (2F$_O$–F$_C$, contoured at 1 σ) of the ligands in the (A) GW9662-bound and (B) SR16832-bound crystal structures of PPARγ LBD with cobound bacterial C9 ligands.
DOI: https://doi.org/10.7554/eLife.43320.020

---

dynamics of PPARγ (*Hughes et al., 2016*; *Hughes et al., 2014*; *Brust et al., 2017*; *Hughes et al., 2012*; *Marciano et al., 2015*). Despite this limitation, chemical shift perturbations were observed for residues near the crystallized C9 binding modes observed in the GW9662 (*Figure 8D*) and SR16832 (*Figure 8E*) crystal structures, including residues on helix 2b/β-sheet surface such as I249, L255, G258, L340, G344, G346, and M348, confirming the crystallized C9 binding modes.

## Cobinding of endogenous UFA and synthetic PPARγ ligands

Our observations described above of fatty acid cobinding were initially made due to the serendipitous retainment of bacterial fatty acids during purification. However, we also showed that addition

**Table 2.** X-ray crystallography data collection and refinement statistics.

| | GW9662 (+ C9) | SR16832 (+ C9) |
|---|---|---|
| Data collection | | |
| Space group | C 1 2 1 | C 1 2 1 |
| Cell dimensions | | |
| $a$, $b$, $c$ (Å) | 92.57, 61.74, 118.38 | 92.61, 62.08, 118.45 |
| $\alpha$, $\beta$, $\gamma$ (°) | 90, 102.15, 90 | 90, 102.34, 90 |
| Resolution | 57.86–2.29 (2.37–2.29) | 45.24–2.73 (2.83–2.73) |
| $R_{pim}$ | 0.029 (0.613) | 0.045 (0.338) |
| I / σ(I) | 11.90 (1.33) | 10.43 (2.00) |
| CC1/2 in highest shell | 0.757 | 0.832 |
| Completeness (%) | 98.94 (98.00) | 85.06 (74.34) |
| Redundancy | 6.6 (6.8) | 1.8 (1.8) |
| | | |
| Refinement | | |
| Resolution (Å) | 2.29 | 2.73 |
| No. of unique reflections | 29660 | 15056 |
| $R_{work}$/$R_{free}$ (%) | 24.9/31.4 | 19.9/28.1 |
| No. of atoms | | |
| Protein | 4145 | 4187 |
| Water | 243 | 60 |
| $B$-factors | | |
| Protein | 33.77 | 30.29 |
| Ligand | 45.19 | 39.84 |
| Water | 30.89 | 21.44 |
| Root mean square deviations | | |
| Bond lengths (Å) | 0.009 | 0.009 |
| Bond angles (°) | 1.02 | 1.09 |
| Ramachandran favored (%) | 95.27 | 90.43 |
| Ramachandran outliers (%) | 1.58 | 1.95 |
| PDB accession code | 6AVI | 6AUG |

*Values in parentheses indicate highest resolution shell.
DOI: https://doi.org/10.7554/eLife.43320.021

of exogenous MCFAs, which are mammalian dietary natural ligands, can indeed cobind with synthetic PPARγ ligands. We therefore wondered whether UFAs such as arachidonic acid (AA) and oleic acid (OA), which are endogenous PPARγ ligands (*Kim et al., 2011*), could also cobind with synthetic ligands. Although these UFAs have longer carbon chains compared to MCFAs, they can bend and bind in more compact conformations.

To explore this hypothesis, we cocrystallized PPARγ LBD with AA or OA and solved two structures to 2.1 and 1.95 Å, respectively (*Table 3*). The structures show two different UFA binding modes in the two chains present in the crystal structure: one where the acid group associates near helix 12 with the hydrophobic fatty acid tail projecting to the back portion of the pocket near the β-sheet surface (*Figure 9A,B*; *Figure 9—figure supplement 1A,B*); and a second where the acid head group adopts a different conformation flipped away from helix 12 with a similar hydrophobic tail conformation near the β-sheet (*Figure 9—figure supplements 1A,B and* and *2A,B*). To determine if a synthetic ligand can cobind with the bound UFA, we soaked GW9662 or rosiglitazone into preformed UFA-bound crystals and were able to solve three crystal structures to ~2.2 Å (*Table 3*), including AA-bound PPARγ with GW9662 (*Figure 9C*; *Figure 9—figure supplement 1C*), OA-bound

**Table 3.** X-ray crystallography data collection and refinement statistics.

| | Arachidonic acid | Oleic acid | GW9662 + Arachidonic acid | GW9662 + Oleic acid | Rosiglitazone + Oleic acid |
|---|---|---|---|---|---|
| Data collection | | | | | |
| Space group | C 1 2 1 | C 1 2 1 | C 1 2 1 | C 1 2 1 | C 1 2 1 |
| Cell dimensions | | | | | |
| $a$, $b$, $c$ (Å) | 93.04, 62.16, 118.96 | 92.93, 62.17, 119.32 | 92.88, 62.10, 119.19 | 92.78, 61.66, 118.63 | 92.83, 61.83, 118.72 |
| $\alpha$, $\beta$, $\gamma$ (°) | 90, 102.38, 90 | 90, 102.20, 90 | 90, 101.90, 90 | 90, 102.15, 90 | 90, 102.34, 90 |
| Resolution | 44.97–2.10 (2.12–2.10) | 38.88–1.95 (2.02–1.95) | 38.87–2.2 (2.279–2.2) | 39.51–2.2 (2.279–2.2) | 57.99–2.24 (2.32–2.24) |
| $R_{pim}$ | 0.039 (0.429) | 0.036 (0.471) | 0.016 (0.277) | 0.014 (0.283) | 0.045 (0.469) |
| I / σ(I) | 10.06 (1.65) | 8.16 (1.34) | 17.06 (2.52) | 17.74 (2.57) | 10.32 (3.02) |
| CC1/2 in highest shell | 0.766 | 0.785 | 0.892 | 0.976 | 0.761 |
| Completeness (%) | 98.42 (95.47) | 95.30 (93.96) | 98.34 (97.39) | 98.24 (97.93) | 97.98 (86.12) |
| Redundancy | 1.9 (1.9) | 1.7 (1.6) | 2.0 (2.0) | 2.0 (2.0) | 3.2 (3.1) |
| | | | | | |
| Refinement | | | | | |
| Resolution (Å) | 2.10 | 1.95 | 2.20 | 2.20 | 2.24 |
| No. of unique reflections | 38363 | 46478 | 33474 | 32923 | 31810 |
| $R_{work}$/$R_{free}$ (%) | 21.3/25.7 | 22.5/27.3 | 22.6/26.2 | 22.5/26.6 | 24.4/28.5 |
| No. of atoms | | | | | |
| Protein | 4102 | 4102 | 4102 | 4118 | 4081 |
| Water | 411 | 502 | 244 | 232 | 224 |
| B-factors | | | | | |
| Protein | 27.08 | 24.94 | 32.75 | 32.18 | 34.00 |
| Ligand | 43.30 | 40.12 | 46.47 | 50.57 | 48.95 |
| Water | 32.32 | 30.69 | 34.42 | 31.61 | 35.11 |
| Root mean square deviations | | | | | |
| Bond lengths (Å) | 0.008 | 0.009 | 0.009 | 0.010 | 0.008 |
| Bond angles (°) | 1.15 | 1.20 | 1.19 | 1.34 | 0.97 |
| Ramachandran favored (%) | 98.19 | 99.00 | 97.59 | 94.82 | 97.18 |
| Ramachandran outliers (%) | 0.60 | 0.20 | 0.40 | 1.99 | 1.01 |
| PDB accession code | 6MCZ | 6MD0 | 6MD2 | 6MD1 | 6MD4 |

[*]Values in parentheses indicate highest resolution shell.

DOI: https://doi.org/10.7554/eLife.43320.025

PPARγ with GW9662 (*Figure 9D*; *Figure 9—figure supplement 1D*), and OA-bound PPARγ with rosiglitazone (*Figure 9E*; *Figure 9—figure supplement 1E*). The structures revealed that the binding of GW9662 or rosiglitazone did not completely displace the bound endogenous UFA (AA or OA). Instead, the binding of the synthetic ligands to UFA-bound PPARγ within the crystals pushed the bound UFA towards the back of the pocket towards the Ω-loop to an alternate site within the crystals (i.e., not the orthosteric binding mode obtained when the UFA was co-crystallized alone), which is apparent in structural overlays of the crystal structures of PPARγ bound to individual UFAs and cobound with GW9662 (*Figure 9—figure supplement 2C,D*).

We performed differential NMR analysis to confirm the ligand cobinding observed in the crystal structures. When PPARγ is bound to GW9662, addition of AA (*Figure 9F*) or OA (*Figure 9G*) caused selective NMR chemical shift perturbations for a similar set of residues affected in the C9 cobinding NMR experiments. Furthermore, NMR analysis showed that addition of OA to rosiglitazone-bound

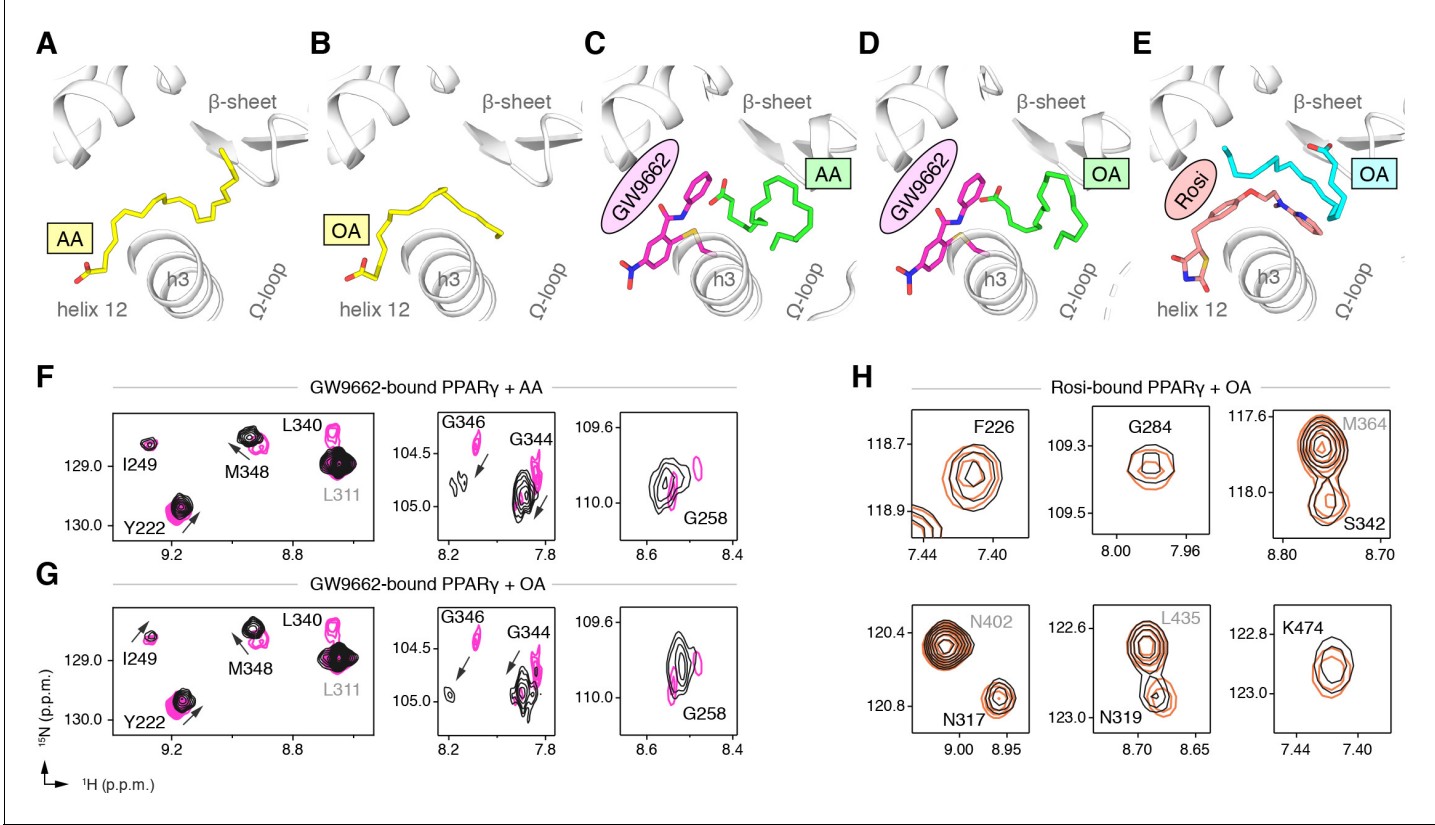

**Figure 9.** Cobinding of endogenous UFAs and synthetic PPARγ ligands. (**A,B**) UFA ligand binding poses in crystal structures of PPARγ LBD (chain A) bound to (**A**) AA and (**B**) OA. (**C–E**) Ligand binding poses in crystal structures obtained from UFA-bound co-crystals soaked with synthetic ligands, including (**C**) AA-bound PPARγ LBD soaked with GW9662, (**D**) OA-bound PPARγ LBD soaked with GW9662, and (**E**) OA-bound PPARγ LBD soaked with rosiglitazone. (**F,G**) Differential 2D [$^1$H,$^{15}$N]-TROSY-HSQC NMR data of PPARγ LBD covalently bound to GW9662 (magenta) confirming cobinding with (**F**) AA (black; one equiv.). and (**G**) OA (black; one equiv.). (**H**) Differential 2D [$^1$H,$^{15}$N]-TROSY-HSQC NMR data of PPARγ LBD bound to rosiglitazone (pink; two equiv.) confirming cobinding with OA (black; one equiv.). Black arrows denote the chemical shift changes in the NMR data. *Also see Figure 9—figure supplements 1 and 2.*

DOI: https://doi.org/10.7554/eLife.43320.022

The following figure supplements are available for figure 9:

**Figure supplement 1.** Omit map (2F$_O$–F$_C$, contoured at 1 σ) of the ligands in crystal structures of PPARγ LBD bound or cobound to (**A**) AA, (**B**) OA, (**C**) GW9662 and AA, (**D**) GW9662 and OA, and (**E**) rosiglitazone and OA.

DOI: https://doi.org/10.7554/eLife.43320.023

**Figure supplement 2.** (**A,B**) UFA ligand binding poses in crystal structures of PPARγ LBD (chain B) bound to (**A**) AA and (**B**) OA.

DOI: https://doi.org/10.7554/eLife.43320.024

PPARγ (*Figure 9H*) also caused selective chemical shift perturbations for residues in close proximity to the cobound OA molecule, including F226, G284, and S342; as well as residues within the AF-2 surface on helix 5 (L317 and K319) and helix 12 (K474).

## Fatty acid cobinding with synthetic covalent ligands affects coactivator interaction

To determine if cobinding of fatty acid and synthetic ligands can affect PPARγ function, we first performed a TR-FRET biochemical assay that enables measurement of a ligand concentration-dependent change in the binding of a peptide derived from the TRAP220 coactivator protein. In this assay format, we reasoned it would be difficult to differentiate orthosteric competition from cobinding using noncovalent synthetic ligands (i.e., TZDs) when titrating MCFAs and UFAs. We therefore performed experiments using PPARγ LBD without and with preincubation of GW9662 or SR16832, the covalent binding synthetic orthosteric ligands we used to structurally observe fatty acid cobinding in our crystal structures and NMR studies above. To test the functional effect of residues implicated in

the C9 cobinding, we generated several mutants. We tested R288A, S342A, and R288A/S342A since these residues directly hydrogen bond via side chain atoms to the cobound C9 ligand in the molecular simulations. Our simulations also revealed a hydrogen bond between C9 and the K265 backbone amide. Because our edaglitazone-bound structure revealed a hydrogen bond between the K265 (Ω-loop) and E291 (helix 3) side chains, which could be important for stabilizing the Ω-loop, we generated K265A and R288A/S342A/K265A mutants to assess the effect of destabilizing the Ω-loop on C9 binding.

In the TR-FRET assay, the overall window of activity is related to the change in binding affinity of the coactivator peptide; a larger increase in the assay window indicates a larger change in affinity. Because coregulator binding affinity is influenced differently by covalent ligands (*Brust et al., 2017*), which causes differences in the basal TR-FRET response (i.e., at low ligand concentrations or DMSO control), we normalized the TR-FRET ratio to easily compare the ligand cobinding and the change in the assay window among the conditions tested.

For wild-type PPARγ, three MCFAs including C9 (*Figure 10A*), decanoic acid (C10; *Figure 10—figure supplement 1A*), and dodecanoic acid (C12; *Figure 10—figure supplement 1B*) and two UFAs AA (*Figure 10B*) and OA (*Figure 10C*) caused a concentration dependent increase in the TRAP220 peptide interaction. When PPARγ was preincubated with GW9662, the fatty acids similarly caused a concentration-dependent increase. However, SR16832 preincubation lowered the assay window of activity to a large degree; smaller MCFAs (C9 and C10) only showed a small increase in TRAP220 peptide interaction, whereas C12 and the endogenous UFAs, AA and OA, did not. This is consistent with our previous report showing that SR16832 more effectively inhibits the UFA docosahexaenoic acid (DHA) from binding to PPARγ compared to GW9662 (*Brust et al., 2017*). Furthermore, the trends observed in this TR-FRET ligand-dependent recruitment assay are similar to peptide affinity data we measured using fluorescence polarization (FP) assays (*Figure 10—figure supplement 2*). Taken together, these data show that MCFAs and UFAs binding to PPARγ when orthosteric pocket is blocked by a covalent ligand can influence PPARγ function.

Compared to wild-type PPARγ or GW9662-bound PPARγ, we found that C9 binding was more potent for SR16832-bound PPARγ (*Figure 10—figure supplement 3*) despite the bulkier addition in SR16832, relative to GW9662, that points towards and blocks the branch II C9 binding mode (*Brust et al., 2017*). Inspection of the SR16832 crystal structure cobound to C9 indicated the increased potency is likely due to hydrogen bonds between the aryl ether/methoxy group of SR16832 with the carboxylate head group and other aliphatic protons in C9 (*Figure 10—figure supplement 4*), the latter of which can form weak hydrogen bonds with methoxy groups (*Palusiak and Grabowski, 2002*). In support of this hydrogen bond network, we performed density functional theory (DFT) calculations, which revealed a net attractive interaction between C9 and the SR16832 methoxy group (10.3 kcal/mol) and a repulsive interaction with the phenyl moiety of GW9662 (8.56 kcal/mol).

We next tested C9 and the UFAs with the mutants that we predicted may impact fatty acid cobinding in the TR-FRET assay, including in the presence of the covalent ligands to isolate the impact of the mutants on alternate site binding. For C9, the R288A mutant (*Figure 10A*) decreased the efficacy (i.e., TR-FRET window) more than the other single mutants without affecting potency to a large degree relative to wild-type PPARγ. However, the R288A/S342A double mutant and the R288A/S342A/K265A triple mutant decreased both the potency and efficacy. For AA (*Figure 10B*) and OA (*Figure 10C*), the potency and efficacy was largely affected in the presence of the covalent ligands with the single mutants, and overall most affected for the double and triple mutants. This indicates these residues are important for driving alternate site affinity and/or functional efficacy of cobound UFAs.

## UFA cobinding synergizes with synthetic TZD ligands to affect coactivator interaction

We used isothermal titration calorimetry (ITC) analysis to determine if there is a functional effect caused by the ligand cobinding scenarios that we captured in our PPARγ crystal structures bound to the noncovalent synthetic TZD ligands edaglitazone (cobound to C9) and rosiglitazone (cobound to OA); and the apparent C9 density present in a previous rosiglitazone-bound structure (PDB: 1FM9) (*Gampe et al., 2000*). Importantly, the nature of ITC experiment allows nearly complete control of the experimental setup, in particular the ability to precisely define the components of the assay (e.

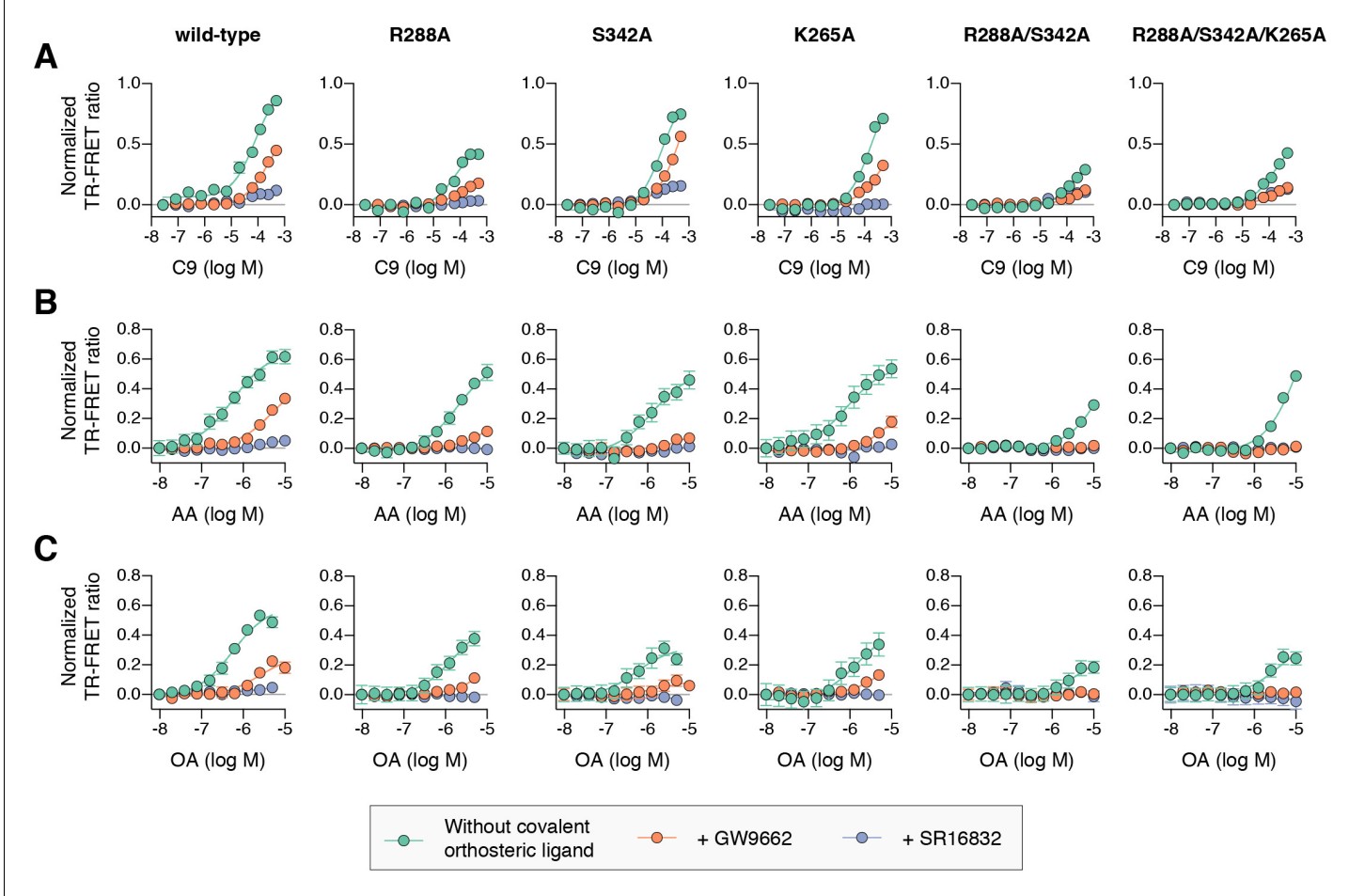

**Figure 10.** Effect of ligand cobinding of a covalent orthosteric synthetic ligand with MCFA or UFA ligands on PPARγ-coactivator recruitment. TR-FRET biochemical assays showing concentration-dependent changes in the recruitment of a peptide derived from the TRAP220 coactivator for (**A**) C9, (**B**) AA, and (**C**) OA. Experiments were performed in the absence or presence of a covalent orthosteric synthetic ligand; and for wild-type PPARγ LBD or mutant variants; as indicated. Data plotted as the average and S.E.M. of three experimental replicates. *Also see Figure 10—figure supplements 1–4.*
DOI: https://doi.org/10.7554/eLife.43320.026

The following figure supplements are available for figure 10:

**Figure supplement 1.** TR-FRET biochemical assays showing concentration-dependent changes in the recruitment of a peptide derived from the TRAP220 coactivator for (**A**) C10 and (**B**) C12.
DOI: https://doi.org/10.7554/eLife.43320.027

**Figure supplement 2.** Fluorescence polarization (FP) assays performed using FITC-labeled peptides derived from (**A**) TRAP220 coactivator and (**B**) NCoR corepressor shows how coregulator affinities (**C**) and (**D**), respectively, are affected by cobinding of synthetic covalent ligands (GW9662 and SR16832) with MCFAs or UFAs.
DOI: https://doi.org/10.7554/eLife.43320.028

**Figure supplement 3.** TRAP220-based TR-FRET assay performed in the absence or presence of covalent antagonists.
DOI: https://doi.org/10.7554/eLife.43320.029

**Figure supplement 4.** Hydrogen bonds/interactions between the SR16832 methoxy group and nonanoic acid.
DOI: https://doi.org/10.7554/eLife.43320.030

g., stoichiometry of UFA and TZD ligands in the sample cell). This is not possible in cellular assays, for example, due to the presence of endogenous cellular ligands that cannot be strictly controlled.

In the ITC experiments, we titrated the TRAP220 coactivator peptide into the sample cell containing PPARγ LBD (*Table 4—source data 1*) to determine how ligand cobinding affected the TRAP220 peptide binding affinity and interaction thermodynamics (*Table 4*). The affinity of the TRAP220 peptide was strengthened in the presence of C9 alone, and more so for OA alone. Inspection of the fitted thermodynamic parameters shows this is driven by a favorable change in entropy (ΔS) in the

**Table 4.** ITC analysis of TRAP220 peptide titrated into PPARγ LBD.

| Ligand | log Ka | Kd (μM) | ΔG (kcal mol⁻¹) | ΔH (kcal mol⁻¹) | TΔs (kcal mol⁻¹) | n (from two replicates) |
|---|---|---|---|---|---|---|
| DMSO | 4.60 (4.53, 4.66) | 25.29 (21.65, 29.77) | −6.271 | −10.53 (−11.58,−9.70) | −4.257 | 0.907, 1.088 |
| C9 (3x) | 5.71 (5.61, 5.82) | 1.94 (1.53, 2.46) | −7.793 | −10.40 (−11.10,−9.81) | −2.612 | 1.059, 1.087 |
| OA (2x) | 5.90 (5.82, 5.97) | 1.27 (1.07, 1.51) | −8.043 | −9.24 (−10.18,−8.31) | −1.192 | 1.276, 1.229 |
| Edaglitazone | 5.90 (5.84, 5.96) | 1.26 (1.10, 1.42) | −8.051 | −10.04 (−10.24,−9.84) | −1.992 | 0.907, 0.863 |
| Rosiglitazone | 5.89 (5.88, 5.94) | 1.28 (1.13, 1.43) | −8.042 | −8.88 (−9.04,−8.72) | −0.838 | 0.907, 0.909 |
| Edaglitazone + C9 | 6.20 (6.10, 6.30) | 0.63 (0.50, 0.78) | −8.457 | −9.71 (−9.98,−9.44) | −1.254 | 0.840, 0.922 |
| Rosiglitazone + C9 | 6.24 (6.16, 6.34) | 0.57 (0.46, 0.69) | −8.523 | −9.12 (−9.38,−8.87) | −0.596 | 0.939, 0.952 |
| Rosiglitazone + OA | 6.16 (6.02, 6.31) | 0.69 (0.49, 0.95) | −8.406 | −7.92 (−8.45,−7.46) | 0.487 | 1.210, 1.258 |

Data represent values from an unbiased global fitting of two independent ITC experiments per condition. The 68.3% confidence interval from global fitting listed as italicized values in parentheses when applicable. Stoichiometry (n value) is listed for each independent experiment. Ligands were present at the following molar equivalents: one equiv. (edaglitazone and rosiglitazone), two equiv. (OA), or three equiv. (C9).

DOI: https://doi.org/10.7554/eLife.43320.031

The following source data is available for Table 4:

**Source data 1.** Thermograms and normalized plotted data from ITC titration of TRAP220 peptide into PPARγ LBD.

Two replicate measurements (green and pink data) per ligand-bound condition (molar equivalents of 1X for rosiglitazone or edaglitazone, 2X for OA, and 3X for C9) were used for the unbiased global ITC analysis.

DOI: https://doi.org/10.7554/eLife.43320.032

fatty acid-bound state. In the presence of edaglitazone or rosiglitazone alone, TRAP220 peptide affinity was similarly strengthen driven by a favorable change in entropy with binding affinities similar to OA-bound PPARγ. In the fatty acid cobound scenarios, a synergetic strengthening of TRAP220 peptide affinity was observed, more than in the presence of each ligand individually, which was also driven by a favorable change in entropy. The entropic-driven nature of the cobinding influence on TRAP220 binding is consistent with our NMR and MD data, which indicate that C9 cobinding with an orthosteric TZD stabilizes the conformation of the AF-2 binding surface. These ITC data confirm that ligand cobinding of a noncovalent synthetic ligand with natural or endogenous ligand affects PPARγ function.

## Ligand cobinding may synergistically affect cellular PPARγ transcription

Given our data above showing structural and functional synergy in the cobinding of enodgenous UFAs and synthetic TZDs to PPARγ, we sought to determine if cobinding could synergistically affect PPARγ transcription in cells. We approached these studies understanding that cobinding of natural/endogenous ligand and a synthetic ligand is difficult to robustly prove in cells. It is not possible to quantitatively control many cellular assay parameters, in particular defining which ligands are present or not in a manner that can be done in the biochemical assays, crystallography, and NMR structural experiments that we performed.

AA and OA, which we used in our structural and biochemical cobinding studies, were among the most abundant UFAs identified using a mass spectrometry metabolomics approach as endogenous ligands present in 3T3-L1 preadipocyte cells and mouse brain and adipose tissue that bind to PPARγ (*Kim et al., 2011*). Our TR-FRET data confirmed that AA and OA bind to the PPARγ LBD and cause recruitment of coactivator peptide, and that alterante site mutants reduce their ability to recruit coactivator. However, when we added exogenous AA and OA to cells, no change in PPARγ

transcription was observed (*Figure 11A*). These observations together indicate that the exogenously added UFAs are already present within cells, which contributes to the basal PPARγ transcriptional activity in the absence of an exogenous ligand; therefore the exogenously added UFAs cause no further increases in PPARγ transcription. Thus, when we performed cell-based transcriptional assays, we considered the results with the premise that PPARγ will be bound to and activated by endogenous ligands present within the cells.

To determine if endogenous ligands within cells may cobind with TZDs and synergistically affect PPARγ transcription, we compared the transcriptional activities of wild-type PPARγ to single, double, and triple mutant variants that showed reduced UFA-induced coactivator recruitment in the TR-FRET biochemical assay. In cells treated with DMSO control, both mutants showed lower basal transcriptional activity (*Figure 11B*) without affecting PPARγ protein levels in cells (*Figure 11—figure supplement 1*) or coactivator affinity (*Figure 11—figure supplement 2*). When considered with our

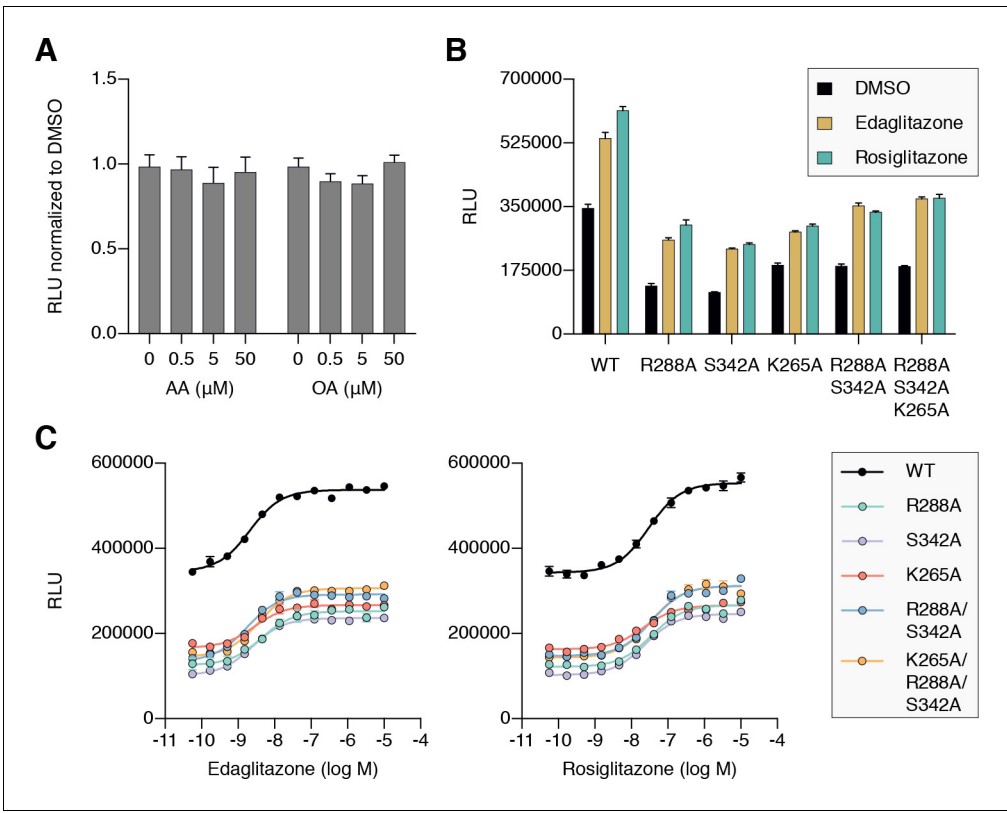

**Figure 11.** Cellular activation of PPARγ by TZDs may be influenced by cobinding of endogenous ligands. (**A**) Luciferase reporter assay measuring full-length wild-type PPARγ transcription in HEK293T cells treated with exogenously added UFAs, which are endogenous ligands present in cells. (**B,C**) Luciferase reporter assay measuring full-length wild-type and mutant PPARγ transcription in HEK293T cells treated with ligand in (**B**) single-point (10 μM) and (**C**) dose-response format. Data plotted as the average and s.e.m. of four experimental replicates; dose response data were fit to a sigmoidal dose response equation. Fitted $EC_{50}$ values are similar for wild-type and the mutant variants for edaglitazone (2–4 nM) and rosiglitazone (22–36 nM). *Also see Figure 11—figure supplement 1* and *2*.

DOI: https://doi.org/10.7554/eLife.43320.033

The following figure supplements are available for figure 11:

**Figure supplement 1.** Western blot analysis protein levels in HEK293T cells transfected with wild-type or mutant full-length PPARγ expression plasmids.

DOI: https://doi.org/10.7554/eLife.43320.034

**Figure supplement 2.** Fluorescence polarization (FP) assays performed using wild-type and mutant PPARγ LBD protein with FITC-labeled peptides derived from (**A**) TRAP220 coactivator and (**B**) NCoR corepressor shows (**C**) coregulator affinities are similar for the mutants.

DOI: https://doi.org/10.7554/eLife.43320.035

biochemical studies, this suggests that the mutants may reduce the binding of and activation by endogenous ligands present in cells.

We next determined how the mutants affected the ability of the synthetic TZD ligands to increase PPARγ transcription. Relative to wild-type PPARγ, the mutants showed reduced activity when cells were treated with saturating synthetic TZD concentrations (*Figure 11B*). The reduction in activity caused by the mutants is not due to decreased TZD affinity because concentration-dependent experiments showed that TZD potency was similar for wild-type PPARγ and the mutants (*Figure 11C*). Furthermore, whereas the overall activation at saturating TZDs levels is lower for the mutants, concentration-dependent relative window of TZD efficacy is similar for wild-type PPARγ and the mutants. When considered with all of the above studies, these data indicate that cobinding of endogenous ligands and synthetic TZDs may synergistically activate PPARγ transcription over the activity of a single ligand alone.

## Discussion

Synthetic ligands and FDA-approved anti-diabetic drugs that target the nuclear receptor PPARγ are thought to compete with and completely displace (i.e., cause unbinding of) natural/endogenous ligands upon binding to an orthosteric pocket to regulate PPARγ activity. This competitive or 'one-for-one' orthosteric ligand exchange model was developed from crystal structures of PPARγ bound to individual ligands, whereby structural overlay of these structures showed that the ligands have overlapping orthosteric binding modes. Our findings here indicate that this classical nuclear receptor ligand exchange model may be more complex than currently appreciated. To our knowledge, this is the first mechanistic description showing that the binding of a synthetic nuclear receptor ligand binding can push aside and cobind with a pre-bound endogenous ligand. This mechanism is particularly evident in our crystallography studies whereby co-crystals of UFA-bound PPARγ, where the UFA crystallized in the orthosteric pocket, was pushed to an alternate site when the crystals were soaked with synthetic ligand. This ligand cobinding mechanism bridges or extends the reach of synthetic ligand within the orthosteric pocket with residues near and within the flexible Ω-loop region that is important in regulating PPARγ structure and function (*Waku et al., 2009a*; *Waku et al., 2009b*; *Jang et al., 2017*; *Bae et al., 2016*; *Puhl et al., 2012*; *Waku et al., 2010*), which is supported by our coregulator interaction assays and particularly our ITC studies that show cobinding of an endogenous UFA and synthetic TZD ligand synergistically improves coactivator affinity compared to each ligand alone. This highlights an important but perhaps underappreciated concept of considering cobinding of natural ligands with designed synthetic ligands in drug discovery efforts.

A growing number of studies have revealed other complex PPARγ ligand binding scenarios. Serotonin and fatty acid metabolites were shown to synergistically bind to different arms of the orthosteric pocket (*Waku et al., 2010*). MCFAs and long-chain oxidized fatty acids were also shown bind in up to three equivalents of the same ligand bound to different arms of the orthosteric pocket (*Itoh et al., 2008*; *Liberato et al., 2012*). We have shown that some synthetic ligands optimized to bind to the orthosteric pocket with high affinity can also bind a second equivalent to the alternate site at lower affinity (*Hughes et al., 2016*; *Li et al., 2008b*; *Hughes et al., 2014*); or one equivalent to the alternate site with an orthosteric antagonist or an oxidized lipid covalently bound to the orthosteric pocket (*Hughes et al., 2014*). Similar to our synthetic ligand alternate binding observations, a crystal structure of PPARγ that showed that a plant-based natural product (luteolin) can bind to the alternate site cobound with a bacterial saturated 14-carbon fatty acid (myristic acid) that copurified during protein expression (*Puhl et al., 2012*). Molecular simulations showed that luteolin bound to the alternate site stabilizes the Ω-loop, and the cobinding is mediated in part through a water bridged hydrogen bond, similar to our cobinding molecular simulations with edaglitazone bound to the orthosteric pocket and C9 bound to the alternate site. In studies on other nuclear receptors, RORγ can cobind a synthetic allosteric ligand and the natural ligand cholesterol in the orthosteric pocket (*Scheepstra et al., 2015*); whereas a study on PXR revealed orthosteric cobinding of a pharmaceutical oestrogen and an organochlorine pesticide (*Delfosse et al., 2015*).

Our observations here, while related to the above studies showing different multiple ligand-bound scenarios, are different. Through a series of crystal structures we mechanistically show that a synthetic PPARγ ligand, including one used as an FDA-approved drug in the treatment of type 2 diabetes, pushes a bound endogenous UFA to an alternate site upon binding rather than

completely displacing it. This likely occurs because the synthetic ligand displays higher affinity than the UFA for the orthosteric pocket, but the UFA also has moderate or reasonable affinity, rather than weak or no affinity, for other portions of the pocket (i.e., the alternate site). The cobound fatty acid in the alternate site, which is located near the β-strand surface and Ω-loop region, is structurally close to two other important surfaces. In a crystal structure of the intact PPARγ/RXRα heterodimer bound to DNA (*Chandra et al., 2008*), there is a long-range interaction between this region in PPARγ and the RXRα DNA-binding domain (DBD). It is possible that ligand cobinding could allosterically affect or synergize with this long-range interaction. Furthermore, the alternate site where the cobound fatty acid is bound is structurally close to a surface containing S245 (S273 in PPARγ isoform 1), which is phosphorylated by the kinase Cdk5 (*Choi et al., 2010*). The antidiabetic efficacy of PPARγ-binding ligands is linked to inhibiting this phosphorylation event (*Choi et al., 2010*; *Choi et al., 2011*), and occupancy of the alternate site can specifically inhibit this phosphorylation event (*Bae et al., 2016*). It is possible that a cobound endogenous ligand at the alternate site may contribute to the antidiabetic efficacy of a synthetic ligand bound to the orthosteric pocket. However, it is difficult to study possible cellular functional effects of synthetic and endogenous ligand cobinding because it is not possible to control the presence of endogenous ligands as one can with synthetic ligands. Our findings may inspire others to develop new techniques or tools to determine how to robustly assess the cellular impact of synthetic and endogenous ligand cobinding.

The ligand cobinding mechanism we discovered here for PPARγ may be broadly applicable to other nuclear receptors and other ligand-binding proteins. For example, many nuclear receptors, including ones not normally thought to be fatty acid responsive, have been shown to interact with and/or be functionally modulated by fatty acids. This list not only includes other PPARs, namely PPARα (*Forman et al., 1997*; *Hostetler et al., 2006*; *Oswal et al., 2013*; *Egawa et al., 2016*) and PPARδ (*Xu et al., 1999*; *Forman et al., 1997*; *Fyffe et al., 2006*), but also receptors for retinoids including RXRα (*Bourguet et al., 2000*; *Egea et al., 2002*; *Lengqvist et al., 2004*; *Xu et al., 2004*) and RAR (*Levi et al., 2015*); steroid hormone receptors including LXRα (*Pawar et al., 2002*; *Caputo et al., 2014*; *Bedi et al., 2017*; *Ducheix et al., 2017*), FXR (*Zhao et al., 2004*), AR (*Mitsuhashi et al., 1988*), TR (*Inoue et al., 1989*; *van der Klis et al., 1991*), ER (*Menendez et al., 2004*; *Moutsatsou et al., 2010*), PR (*Kato et al., 1987*), and GR (*Kato et al., 1987*; *Viscardi and Max, 1993*) including a second non-hormone site in GR (*Vallette et al., 1991*); as well as orphan receptors including HNF4α (*Dhe-Paganon et al., 2002*; *Wisely et al., 2002*), Nurr1 (*de Vera et al., 2016*; *de Vera et al., 2018*), Nur77 (*Vinayavekhin and Saghatelian, 2011*), and TR4 (*Xie et al., 2009*). It is possible that synthetic ligands targeting these nuclear receptors also cobind with fatty acids, or even other types of endogenous ligands, with synergistic affects, positive or negative, on receptor function. In fact, fatty acids are considered to the ancestral nuclear receptor ligands (*Sumida, 1995*) and also interact with and regulate the function of a variety of other protein targets including kinases (*McPhail et al., 1984*; *Murakami and Routtenberg, 1985*; *Khan et al., 1995*), GPCRs (*Stoddart et al., 2008*; *Yonezawa et al., 2013*), and ion channels (*Ordway et al., 1991*; *Antollini and Barrantes, 2016*), which are targets (or putative targets) of synthetic ligands and clinically used drugs. More broadly, fatty acids and lipids are just one type of endogenous metabolite present in cells and it is estimated that most proteins likely bind endogenous metabolites (*Lindsley and Rutter, 2006*; *Gallego et al., 2010*; *Li et al., 2010*; *Li and Snyder, 2011*; *Orsak et al., 2012*; *Link et al., 2013*). Thus, the concept of synthetic ligand/drug cobinding with endogenous metabolites to targets may be more likely than not.

## Materials and methods

### Materials and reagents

Rosiglitazone, edaglitazone, and GW9662 ligands were obtained from commercial sources (Cayman and Tocris Bioscience). SR16832 was synthesized in-house as previously described (*Brust et al., 2017*). Peptides of LXXLL-containing motifs from TRAP220 (residues 638–656; NTKNHPMLMNLLKD NPAQD) and NCoR (2256–2278; DPASNLGLEDIIRKALMGSFDDK) containing a N-terminal FITC label with a six-carbon linker (Ahx) and an amidated C-terminus for stability were synthesized by LifeTein.

## Protein purification

Human PPARγ LBD (residues 203–477, isoform one numbering; or residues 231–505, isoform two numbering) was expressed in *E. coli* BL21(DE3) cells using autoinduction ZY media or M9 minimal media supplemented with NMR isotopes as a Tobacco Etch Virus (TEV)-cleavable N-terminal hexahistidine (6xHis)-tagged fusion protein using a pET46 Ek/LIC vector (Novagen) and purified using nickel affinity chromatography and size exclusion chromatography. The purified proteins were concentrated to 10 mg/mL in a buffer consisting of 20 mM potassium phosphate (pH 7.4), 50 mM potassium chloride, 5 mM tris(2-carboxyethyl)phosphine (TCEP), and 0.5 mM ethylenediaminetetra-acetic acid (EDTA). Purified protein was verified by SDS-PAGE as >95% pure. For studies using a covalent orthosteric antagonist, PPARγ LBD protein was incubated with at least a ~ 1.05 x excess of GW9662 or SR16832 at 4°C for 24 hr to ensure covalent modification to residue C285, then buffer exchanged the sample to remove excess covalent antagonist and DMSO. Complete attachment of the covalent antagonist occurs within 30–60 min, as detected using ESI-MS with a LTQ XL linear Ion trap mass spectrometer (Thermo Scientific). For studies using delipidated protein, two protocols were used (LIPIDEX and refolding) that gave similar results. For LIPDEX delipidation, protein was incubated with LIPIDEX-1000 (Perkin Elmer) at 37°C for 1 hr and buffer exchanged into the afore-mentioned phosphate buffer. For delipidation by refolding, protein was first subjected to a chloro-form/methanol lipid extraction (*Bligh and Dyer, 1959*) and the denatured protein was refolded using a fast dilution/dialysis procedure followed by size exclusion chromatography (*Armstrong et al., 2014*). NMR spectra of delipidated protein obtained from both procedures were essentially identical. In some experiments (i.e., FP and ITC), we used nondelipidated protein sub-jected to column chromatography purification that significantly reduced the amount of (putatively) bound bacterial lipids. This was verified by NMR as well as ITC and FP experiments where the addi-tion of C9 or any other fatty acid (e.g., AA or OA) to the protein caused a strengthening of the coac-tivator binding affinity vs. apo-protein as detected by ITC; no change in affinity would be expected if the protein was bound/saturated by bacterial lipid.

## TR-FRET ligand displacement and coregulator interaction assays

The time-resolved fluorescence resonance energy transfer (TR-FRET) assays were performed in black low-volume 384-well plates (Greiner). For the ligand displacement assay, each well contained 1 nM delipidated 6xHis-PPARγ LBD protein, 1 nM LanthaScreen Elite Tb-anti-HIS Antibody (Thermo Fisher Scientific), and 5 nM Fluormone Pan-PPAR Green (Invitrogen) in TR-FRET buffer containing 20 mM potassium phosphate (pH 8), 50 mM potassium chloride, 5 mM TCEP, and 0.005% Tween-20 in the absence or presence of ligand in triplicate at a concentration equal to the highest protein concentra-tion to ensure complete formation of ligand-bound protein. For the TR-FRET coregulator assay, each well contained 4 nM 6xHis-PPARγ LBD protein (with or without covalent modification by a covalent antagonist), 1 nM LanthaScreen Elite Tb-anti-HIS Antibody (Thermo Fisher Scientific), and 400 nM FITC-labeled TRAP220 or NCoR peptide described above in TR-FRET buffer. Ligand stocks were prepared via serial dilution in DMSO, added to wells in triplicate, and plates were incubated at 25°C for 1 hr and read using BioTek Synergy Neo multimode plate reader. The Tb donor was excited at 340 nm; its fluorescence emission was monitored at 495 nm, and the acceptor FITC emission was measured at 520 nm; and the TR-FRET ratio was calculated as the signal at 520 nm divided by the signal at 495 nm. Data were plotted using GraphPad Prism as TR-FRET ratio vs. ligand concentra-tion; TR-FRET coregulator data were fit to sigmoidal dose response curve equation, and ligand dis-placement data were fit to the one site – Fit $K_i$ binding equation to obtain $K_i$ values using the known binding affinity of Fluormone Pan-PPAR Green (2.8 nM; Invitrogen PV4894 product insert).

## Fluorescence polarization coregulator interaction assays

Wild-type or mutant 6xHis-PPARγ LBD in assay buffer (20 mM potassium phosphate (pH 8), 50 mM potassium chloride, 5 mM TCEP, 0.5 mM EDTA, and 0.01% Tween-20) was diluted by serial dilution and plated with 200 nM FITC-labeled TRAP220 or NCoR peptides described above in black 384-well plates (Greiner). The plate was incubated at 4°C for 1 hr, and fluorescence polarization was mea-sured on a BioTek Synergy Neo multimode plate reader at 485 nm emission and 528 nm excitation wavelengths. Data were plotted using GraphPad Prism as fluorescence polarization signal in millipo-larization units vs. protein concentration and fit to a one site total binding equation.

## Luciferase reporter assays and western analysis of protein levels

For the luciferase reporter assay, HEK293T cells (obtained from ATCC; cat# CRL-3216; RRID:CVCL_0063; authenticated by morphology and had no evidence of mycoplasma contamination) were cultured in DMEM media (Gibco) supplemented with 10% fetal bovine serum (FBS) and 50 units mL$^{-1}$ of penicillin, streptomycin, and glutamine and grown to 90% confluency in a T-75 flask before seeding 4 million cells per well in 10 cm dishes. Seeded cells were transfected with 4.5 µg of pCMV6-XL4 plasmid containing full-length human PPARγ2 and 4.5 µg of 3X-multimerized PPAR response element (PPRE)-luciferase reporter plasmid with 27 µL of X-treme Gene nine transfection reagent (Roche) in Opti-MEM reduced serum media (Gibco). Following 18 hr incubation at 37°C in a 5% CO$_2$ incubator, transfected cells were plated in white 384-well plates (Perkin Elmer) at a density of 10,000 cells per well (20 µL volume). After plating, cells were incubated 4 hr before treatment in quadruplicate with 20 µL of either vehicle control (2% DMSO in DMEM media) or 1:3 serial dilution of ligand (edaglitazone or rosiglitazone) from 2 pM to 630 nM; DMSO concentration was constant at 2% in all wells. Following a second 18 hr incubation, treated cells were developed with 20 µL Britelite Plus (Perkin Elmer), and luminescence was measured using a BioTek Synergy Neo multimode plate reader. Data were plotted in GraphPad Prism as luminescence vs. ligand concentration and fit to a sigmoidal dose response curve.

For the western analysis, HEK293T cells (obtained from ATCC; cat# CRL-3216; RRID:CVCL_0063; authenticated by morphology and had no evidence of mycoplasmia contamination) were transfected described above except in 6-well plates (Corning) with 250,000 cells and 1 µg of pCMV6-XL4 plasmid containing wild-type or mutant full-length human PPARγ2 with 3 µL of X-treme Gene nine and incubated for 48 hr. Transfected cells were then lysed in TNT buffer (150 mM NaCl, 0.1 M Tris, 1% Triton-X 100, pH 7.5) and incubated for 1 hr at 4 ˚C. Protein concentration was determined by BCA assay (ThermoFisher). Protein (20 µg) was loaded onto 4–15% gradient gels (Bio-Rad) and wet transferred onto PVDF. The membrane was blocked for 1 hr at room temperature with 5% milk in PBS. All antibodies were diluted in 1% milk in PBS. After blocking, the membrane was incubated overnight at 4 ˚C with 1:1000 dilution of rabbit anti-PPARγ (Cell Signaling Technology, #2435; RRID:AB_2166051). The following day, the blot was washed with PBST and treated with 1:3000 dilution of anti-rabbit IgG-HRP (Cell Signaling Technology, #7074; RRID:AB_2099233) for 45 min at room temperature, then washed with PBST and developed with ECL (ThermoFisher). The membrane was stripped, blocked, and re-probed with 1:1000 mouse anti-actin (EMD/MilliporeSigma, MAB1501; RRID:AB_2223041) overnight at 4 ˚C. The following day the blot was washed with PBST and treated with 1:10,000 dilution of anti-mouse IgG-HRP (Sigma Aldrich, #A9917; RRID:AB_258476) for 45 min at room temperature, then washed with PBST and developed with ECL (Thermo Fisher).

## Crystallization, structure determination, and structural analyses

PPARγ LBD protein was purified as described above. For edaglitazone, arachidonic acid, oleic acid and GW9662-PPARγ LBD complexes, protein was incubated at a 1:3 protein/ligand molar ratio in PBS overnight, concentrated to 10 mg/mL, and buffer exchanged. Crystals were obtained after 3–5 days at 22°C by sitting-drop vapor diffusion against 50 µL of well solution using 96-well format crystallization plates. The crystallization drops contained 1 µL of protein/ligand sample mixed with 1 µL of reservoir solution containing 0.1 M Tris-HCl (pH 7.6) and 0.8 M sodium citrate. Crystals for the GW9662/AA, GW9662/OA, rosiglitazone/OA PPARγ LBD complexes were obtained by soaking GW9662 or rosiglitazone (1 mM in reservoir solution containing 5% DMSO) into preformed AA or OA bound PPARγ LBD crystals. Crystals for the SR16832-PPARγ LBD complex were obtained via 'dry' co-crystallization method (*Fyffe et al., 2006*) after ~14 days using a reservoir solution containing 0.1 M MOPS (pH 7.6) and 0.8 M sodium citrate. All crystals were flash-frozen in liquid nitrogen before data collection. Data collection for the edaglitazone-bound structure was carried out at LS-CAT Beamline 21-ID-D at the Advanced Photon Source at Argonne National Laboratory; the GW9662-bound structures at Beamline 5.0.2 at Berkeley Center for Structural Biology; and the arachidonic acid, oleic acid and SR16832-bound structures using our home source MicroMax007 HF x-ray generator equipped with the mar345 detector. Data were processed, integrated, and scaled with the programs Mosflm (*Battye et al., 2011*) and Scala in CCP4 (94). The structure was solved by molecular replacement using the program Phaser (*McCoy et al., 2007*) that is implemented in the Phenix package (*Adams et al., 2010*) using a previously published PPARγ LBD structure (PDB:

1PRG) (*Nolte et al., 1998*) as the search model. The structure was refined using Phenix with several cycles of interactive model rebuilding in Coot (*Emsley and Cowtan, 2004*). Figures were prepared using PyMOL (Schrödinger), and the orthosteric pocket and alternate site volumes shown in *Figure 1* were calculated using HOLLOW (*Ho and Gruswitz, 2008*).

## C9 fitting into published ligand-bound PPARγ crystal structures

We analyzed several high-resolution crystal structures, including structures where based on sequence information it was possible that Ω-loop density was present, to determine if density consistent with a medium chain fatty acid could be present. Our analysis included PDB codes 1fm6, 1fm9, 1k74, 1zgy, 2hfp, 2prg, 2q5s, 2q61, 2q6s, 2q6r, 2zk1, 2zk2, 2zk3, 2zk4, 2zk5, 2zk6, 2zvt, 3adw, 3b0q, 3dzu, 3dzy, 3e00, 3qt0, 3r8i, 3t03, 4ema, 4fgy, 4xld, 5two. In some cases it was assumed that the potential fatty acid electron density may have been fitted by water molecules in the previous structure determinations near the site where we observed a MCFA in our edaglitazone-bound structure. In other cases, we analyzed the electron density present in mtz files downloaded from the PDBe (e.g., http:// www.ebi.ac.uk/pdbe/coordinates/files/xxxx_map.mtz; where xxxx is the PDB code in lower case letters) for unmodeled density. After removal of all water molecules from the PDB structure file, a ligand search followed by interactive real space refinement was performed using the crystallography atomic modeling program Coot (*Emsley and Cowtan, 2004*); this structure was subjected to refinement with rebuilt water molecules using Phenix (*Adams et al., 2010*). Of the structures analyzed, C9 was successfully fitted into the electron density of PPARγ LBD bound to the TZD ligands rosiglitazone (PDB: 1FM6) and MCC-555 (PDB: 3B0Q), as well as the non-TZD ligands SR145 (PDB: 2Q61) and GQ-16 (PDB: 3T03).

## NMR spectroscopy

Two-dimensional (2D) [$^1$H,$^{15}$N]-transverse relaxation optimized spectroscopy (TROSY)-heteronuclear single quantum correlation (HSQC) data were collected at 298K using a Bruker 700 Mhz NMR instrument equipped with a QCI cryoprobe. Samples contained approximately 200 μM protein in a NMR buffer containing 50 mM potassium phosphate (pH 7.4), 20 mM potassium chloride, 1 mM TCEP, and 10% D$_2$O. Data were processed and analyzed using Topspin 3.0 (Bruker) and NMRViewJ (One-Moon Scientific, Inc.) (*Johnson, 2004*). NMR analysis was performed using previously described rosiglitazone-bound NMR chemical shift assignments (BMRB entry 17975) (*Hughes et al., 2012*); for the analysis of PPARγ bound to edaglitazone and/or C9, the analysis was limited to well resolved peaks for residues with chemical shift values similar to PPARγ bound to rosiglitazone via the minimum chemical shift procedure (*Williamson, 2013*).

## Molecular dynamics simulations

Edaglitazone was modeled with the NH in the thiazolidinedione group (pKa ~6.4 (101)) as deprotonated, thus with a total charge of −1 at physiologic pH. C9 (pKa of the carboxyl group is ~5) was modeled as deprotonated and having an overall −1 charge. Appropriate protons were added to edaglitazone and C9 structures extracted from the crystal structure using Chimera (*Pettersen et al., 2004*) and submitted to REDD server (*Dupradeau et al., 2010*). The resulting mol2 files were used in tleap (AMBER version 16) along with the crystal structure bound to edaglitazone and C9 (PDB: 5UGM). Missing amino acids were added via Modeller (*Sali and Blundell, 1993*) within Chimera, including an N-terminal glycine that was added to match the NMR construct (vestige of tag cleavage). The structures (without ligand) were run through the H ++ server (*Anandakrishnan et al., 2012*) to calculate probable protonation states of titratable protons based on structure. Ligands were then added to the H ++ PDB file via overlay of the crystal structure in Chimera. Pdb4amber was used to generate a PDB file with amber names and antechamber was used to give gaff2 names to the ligand mol2 files. Parmchk2 was used to generate force modification files for the ligands with gaff2 parameters. Sodium ions were used to neutralize the ligand-protein complex, and KCl was added to 50 mM (to match the NMR experimental conditions) using TIP3P specific parameters (*Joung and Cheatham, 2009*). The system was solvated with TIP3P (*Jorgensen et al., 1983*) water in a truncated octahedron box at least 10 Å from any protein or ligand atom. FF14SB (*Maier et al., 2015*) was used for protein parameters. This build was then minimized and equilibrated in nine steps with restraints on heavy atoms decreasing with each step with the last minimization and equilibration

steps without restraints. Production runs were then carried out with hydrogen mass repartitioned (*Hopkins et al., 2015*) parameter files to enable four fs time steps. Constant pressure replicate production runs were carried out with independent randomized starting velocities. Pressure was controlled with a Monte Carlo barostat and a pressure relaxation time (taup) of 2ps. Temperature was kept constant at 310 K with Langevin dynamics utilizing a collision frequency (gamma_ln) of 3 ps$^{-1}$. The particle mesh ewald method was used to calculate non-bonded atom interactions with a cutoff (cut) of 8.0 Å. SHAKE (*Ryckaert et al., 1977*) was used to allow longer time steps in addition to hydrogen mass repartitioning. Analysis of trajectories was performed using cpptraj (*Roe and Cheatham, 2013*) and pytraj. Hydrogen bond analysis was performed using dist = 3.5 Å and angle = 100° (*Fabiola et al., 2002*).

## Density functional theory calculations

All calculations were performed at B3LYP/6–311 + G (*Perdew et al., 1996*; *Parr and Yang, 1989*) level using Gaussian 98 package. In order to reduce the calculation time/computational cost, we used a simplified model by only considering the C9-interacting parts of GW9662 (phenyl group), SR16832 (6-methoxyquinoline group) in the calculations. Geometry optimizations were performed for GW9662, SR16832, C9, GW9662 + C9, and SR16832 + C9 using the structural conformations present in the crystal structures.

## Isothermal titration calorimetry

ITC experiments were carried out on a MicroCal iTC200 calorimeter (GE/MicroCal, Northampton, MA) using the iTC200 software (v 1.24.2) for instrument control and data acquisition. PPARγ LBD was present in the sample cell that also contained either 1.5% DMSO (control) or ligand with the weaker binding natural ligand (C9) or endogenous ligand (OA) added in 2 or three molar equiv., respectively, relative to the high affinity synthetic ligands (edaglitazone or rosiglitazone) for the following conditions — C9 (three molar equiv.), OA (two molar equiv.), rosiglitazone (one molar equiv.), edaglitazone (one molar equiv.) — that were diluted to 60 μM in reaction buffer consisting of 20 mM KPO4 (pH 7.4), 50 mM KCl, 5 mM TCEP, 0.5 mM EDTA, and 0.5%DMSO. TRAP220 peptide powder was dissolved in the same buffer at 600 μM. The reaction cell contained 60 μM protein-ligand complex (200 μl) was titrated with 0.4 μL pre-injection followed by nineteen 2 μL injections of 600 μM TRAP220. Mixing was carried out at 25°C with reference power and rotational stirring set at 5 μcal/s and 1200 rpm, respectively. Data were processed in NITPIC (*Keller et al., 2012*) to determine binding stoichiometry and further analyzed by unbiased global fitting of two replicate runs per ligand-bound condition in SEDPHAT (*Zhao et al., 2015*; *Brautigam et al., 2016*), followed by export to GUSSI for publication-quality figure preparation (*Brautigam, 2015*). The SEDPHAT fitting model used was A + B to AB heteroassociation and the fit parameters were enthalpy (ΔH) and affinity (K$_d$).

# Acknowledgements

This work was supported in part by National Institutes of Health (NIH) grants R01DK101871 (DJK), R00DK103116 (TH), and F32DK108442 (RB); American Heart Association (AHA) fellowship award 16POST27780018 (RB); National Science Foundation (NSF) funding to the Summer Undergraduate Research Fellows (SURF) program at The Scripps Research Institute [Grant 1659594]; and the Academic Year Research Internship for Undergraduates (AYRIU) program at The Scripps Research Institute.

# Additional information

### Funding

| Funder | Grant reference number | Author |
| --- | --- | --- |
| National Institute of Diabetes and Digestive and Kidney Diseases | R01DK101871 | Douglas J Kojetin |
| American Heart Association | 16- POST27780018 | Richard Brust |

| National Science Foundation | 1659594 | Sarah A Mosure |
| The Scripps Research Institute | | Sarah A Mosure |
| National Institute of Diabetes and Digestive and Kidney Diseases | R00DK103116 | Travis S Hughes |
| National Institute of Diabetes and Digestive and Kidney Diseases | F32DK108442 | Richard Brust |

The funders had no role in study design, data collection and interpretation, or the decision to submit the work for publication.

### Author contributions
Jinsai Shang, Conceptualization, Formal analysis, Investigation, Visualization, Writing—review and editing; Richard Brust, Sarah A Mosure, Jared Bass, Paola Munoz-Tello, Investigation, Writing—review and editing; Hua Lin, Resources, Investigation, Writing—review and editing; Travis S Hughes, Resources, Writing—review and editing; Miru Tang, Formal analysis, Investigation, Writing—review and editing; Qingfeng Ge, Supervision, Investigation, Writing—review and editing; Theodore M Kamenecka, Resources, Supervision, Writing—review and editing; Douglas J Kojetin, Conceptualization, Formal analysis, Supervision, Funding acquisition, Investigation, Visualization, Writing—original draft, Project administration, Writing—review and editing

### Author ORCIDs
Jinsai Shang (iD) http://orcid.org/0000-0001-8164-1544
Richard Brust (iD) http://orcid.org/0000-0002-9200-1101
Sarah A Mosure (iD) http://orcid.org/0000-0003-4477-5396
Paola Munoz-Tello (iD) http://orcid.org/0000-0002-5225-1907
Travis S Hughes (iD) http://orcid.org/0000-0002-5764-5884
Douglas J Kojetin (iD) http://orcid.org/0000-0001-8058-6168

### Decision letter and Author response
Decision letter https://doi.org/10.7554/eLife.43320.076
Author response https://doi.org/10.7554/eLife.43320.077

## Additional files

### Supplementary files
• Transparent reporting form
DOI: https://doi.org/10.7554/eLife.43320.036

### Data availability
Crystal structures and diffraction data have been deposited in the PDB under accession codes 5UGM, 6AVI, 6AUG, 6MCZ, 6MD0, 6MD1, 6MD2, and 6MD4.

The following datasets were generated:

| Author(s) | Year | Dataset title | Dataset URL | Database and Identifier |
|---|---|---|---|---|
| Shang J, Kojetin DJ | 2017 | Crystal Structure of Human PPARgamma Ligand Binding Domain in Complex with Edaglitazone | http://www.rcsb.org/structure/5UGM | Protein Data Bank, 10.2210/pdb5UGM/pdb |
| Shang J, Kojetin DJ | 2017 | Crystal Structure of Human PPARgamma Ligand Binding Domain in Complex with GW9662 and Nonanoic acid | http://www.rcsb.org/structure/6AVI | Protein Data Bank, 10.2210/pdb6AVI/pdb |
| Shang J, Kojetin DJ | 2017 | Crystal Structure of Human PPARgamma Ligand Binding | http://www.rcsb.org/structure/6AUG | Protein Data Bank, 10.2210/pdb6AUG/ |

| | | | | |
|---|---|---|---|---|
| | | Domain in Complex with SR16832 | | pdb |
| Shang J, Kojetin DJ | 2018 | Crystal Structure of Human PPARgamma Ligand Binding Domain in Complex with Arachidonic Acid | http://www.rcsb.org/structure/6MCZ | Protein Data Bank, 6MCZ |
| Shang J, Kojetin DJ | 2018 | Crystal Structure of Human PPARgamma Ligand Binding Domain in Complex with Oleic Acid | http://www.rcsb.org/structure/6MD0 | Protein Data Bank, 6MD0 |
| Shang J, Kojetin DJ | 2018 | Crystal Structure of Human PPARgamma Ligand Binding Domain in Complex with GW9662 and Oleic acid | http://www.rcsb.org/structure/6MD1 | Protein Data Bank, 6MD1 |
| Shang J, Kojetin DJ | 2018 | Crystal Structure of Human PPARgamma Ligand Binding Domain in Complex with GW9662 and Arachidonic acid | http://www.rcsb.org/structure/6MD2 | Protein Data Bank, 6MD2 |
| Shang J, Kojetin DJ | 2018 | Crystal Structure of Human PPARgamma Ligand Binding Domain in Complex with Rosiglitazone and Oleic acid | http://www.rcsb.org/structure/6MD4 | Protein Data Bank, 6MD4 |

The following previously published datasets were used:

| Author(s) | Year | Dataset title | Dataset URL | Database and Identifier |
|---|---|---|---|---|
| Kojetin D, Johnson B | 2012 | PPARgamma LBD complexed with rosiglitazone | http://www.bmrb.wisc.edu/data_library/summary/?bmrbId=17975 | Biological Magnetic Resonance Bank, 10.13018/BMR17975 |
| Liberato MV, Nascimento AS, Polikarpov I | 2012 | Human PPAR gamma in complex with nonanoic acids | https://www.rcsb.org/structure/4EM9 | Protein Data Bank, 10.2210/pdb4EM9/pdb |
| Liberato MV, Nascimento AS, Polikarpov I | 2012 | Human peroxisome proliferator-activated receptor gamma in complex with rosiglitazone | https://www.rcsb.org/structure/4EMA | Protein Data Bank, 10.2210/pdb4EMA/pdb |
| Wang Z | 2009 | Crystal Structure of PPARg in complex with T2384 | https://www.rcsb.org/structure/3K8S | Protein Data Bank, 10.2210/pdb3K8S/pdb |
| Nolte RT, Wisely GB, Milburn MV | 1998 | Ligand-binding domain of the human peroxisome proliferator activated receptor gamma | https://www.rcsb.org/structure/2PRG | Protein Data Bank, 10.2210/pdb2PRG/pdb |
| Gampe Jr RT, Montana VG, Lambert MH, Miller AB, Bledsoe RK, Milburn MV, Kliewer SA, Willson TM, Xu HE | 2000 | The 2.1 angstrom resolution crystal structure of the heterodimer of the human rxralpha and ppargamma ligand binding domains respectively bound with 9-cis retinoic acid and rosiglitazone and co-activator peptides | https://www.rcsb.org/structure/1FM6 | Protein Data Bank, 10.2210/pdb1FM6/pdb |
| Tomioka D, Hashimoto H, Sato M, Shimizu T | 2011 | Human PPAR gamma ligand binding domain in complex with MCC555 | https://www.rcsb.org/structure/3B0Q | Protein Data Bank, 10.2210/pdb3B0Q/pdb |
| Rajagopalan S, Webb P, Baxter JD, Brennan RG, Phillips KJ | 2011 | Crystal structure of PPAR gamma ligand binding domain in complex with a novel partial agonist GQ-16 | https://www.rcsb.org/structure/3T03 | Protein Data Bank, 10.2210/pdb3T03/pdb |
| Bruning JB, Nettles KW | 2007 | Crystal Structure of PPARgamma ligand binding domain bound to partial agonist SR145 | https://www.rcsb.org/structure/2Q61 | Protein Data Bank, 10.2210/pdb2Q61/pdb |
| Tomioka D, Hashimoto H, Sato M, Shimizu T | 2011 | Human PPAR gamma ligand binding dmain complexed with GW9662 in a covalent bonded form | https://www.rcsb.org/structure/3B0R | Protein Data Bank, 10.2210/pdb3B0R/pdb |
| Nolte RT, Wisely GB, Milburn MV | 1998 | Ligand binding domain of the human peroxisome proliferator activated receptor gamma | https://www.rcsb.org/structure/1PRG | Protein Data Bank, 10.2210/pdb1PRG/pdb |

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
