## [Decision Letter]

[Editors’ note: a previous version of this study was rejected after peer review, but the authors submitted for reconsideration. The first decision letter after peer review is shown below.]

Thank you for submitting your work entitled "Coincident binding of synthetic and natural ligands to the nuclear receptor PPARγ" for consideration by *eLife*. Your article has been reviewed by three peer reviewers, and the evaluation has been overseen by a Senior/Reviewing Editor. The following individuals involved in review of your submission have agreed to reveal their identity: Kevin H Gardner (Reviewer #2). Two other reviewers remain anonymous.

This manuscript by Kojetin et al. presents a high-quality and interdisciplinary analysis of ligand binding to PPAR γ, specifically exploring the roles of co-binding artificial ligands to the traditionally-targeted orthosteric site in combination with natural (fatty acid) ligands at an adjacent alternate site. Working together, these two sets of ligands cooperate to impact the structure and function of PPARg segments involved in coactivator/coregulatory recruitment, with effects on in vitro and cellular activities.

The authors document their findings through a combination of structural biology, biophysics and biochemical approaches. The reviewers are impressed by the crystal structures showing co-occupancy of PPARg co-bound to the agonist edaglitazone and a nonanoic (C9) fatty acid at adjacent sites, with a concomitant ordering of the omega loop/AF2 surface involved in coactivator recruitment. These results are coupled with a retrospective analysis of other published PPARg/agonist structures identifying previously-nonmodeled fatty acid densities and MD simulations showing linkage between ligand binding and omega loop rigidification (Figure 8). They also show that mutation of the nonanoic acid binding site reduces PPARg activity in cotransfection assay, demonstrating the importance of the integrity of this site for the receptor's basal and synthetic ligand activity.

While recognizing the quality of the work and its thoroughness, the principal concern regarding further review at *eLife* are issues of novelty and the level of new conceptual advance, particularly given the depth of information already available for this class of receptors. The idea that a natural ligand can synergize with a synthetic ligand to activate the receptor is not rigorously supported by the data in this paper. A number of structural studies have already convincingly demonstrated the ability of nuclear receptors to be occupied by multiple ligand species (the authors appropriately cited several of these). Moreover, this is not the first structure to report that nonanoic acid can be co-bound to PPARg with other similar ligands (PLoS One 7:e36297, 2012). Prior papers (Hughes et al., 2014 and Puhl et al., 2012 discussed in the third paragraph of the Discussion) detail a mix of natural and artificial compounds binding concurrently to PPARg's orthosteric pocket and the alternative site, presenting structural and cell biology data to support the nature of the interaction and functional relevance. Finally, beyond PPARg, it is not clear how broadly this concept might apply.

Specific points raised by the reviewers are appended below, which may be helpful in revising the manuscript for submission elsewhere.

*Reviewer #1:*

This paper reports structural studies of PPARg bound to a TZD with the fortuitous finding that a bacterial lipid (nonanoic acid) is co-bound in one of the cavities of the ligand binding pocket. The authors provide a series of well-conceived structural studies to support the co-binding concept. They also show that mutation of the nonanoic acid binding site reduces PPARg activity in cotransfection assay, demonstrating the importance of the integrity of this site for the receptor's basal and synthetic ligand activity.

The concern of the paper is novelty and impact. A number of structural studies have already convincingly demonstrated the ability of nuclear receptors to be occupied by multiple ligand species (the authors appropriately cited several of these). Moreover, this is not the first structure to report that nonanoic acid can be co-bound to PPARg with other similar ligands (PLoS One 7:e36297, 2012). Perhaps the key novelty of the paper is idea that a natural ligand can synergize with a synthetic ligand to activate the receptor, but this concept was not rigorously supported by the data in this paper. Finally, beyond PPARg, it is not clear how broadly this concept might apply.

Other specific comments:

1) The structure reveals the bacterial lipid, nonanoic acid, bound to PPARg, a finding that has been shown by others as the authors note. However, this finding appears to be an artifact of preparing the protein in bacteria, an artifact that has been observed in numerous other nuclear receptor structures. Importantly, however, no evidence is given to support the idea that nonanoic acid is a naturally occurring ligand at relevant concentrations in mammalian cells. The authors also show one set of data using C10 and C12 lipids that can bind competitively to the pocket, but no data were presented to show they function similarly to the C9 lipid.

2) The data in Figure 10 might support the idea of endogenous cellular lipids bound to PPARg, but again this was not proven and there could be other interpretations of these results. These mutations may simply compromise basal activity due to non-ligand dependent changes in the structure of the pocket, similar to that revealed in other nuclear receptors.

*Reviewer #2:*

This manuscript by Kojetin et al. provides an interdisciplinary look at ligand binding to PPAR γ, specifically exploring the roles of co-binding artificial ligands to the traditionally-targeted orthosteric site in combination with natural (fatty acid) ligands at an adjacent alternate site. Working together, these two sets of ligands cooperate to impact the structure and function of PPARg segments involved in coactivator/coregulatory recruitment, with predictable effects on in vitro and cellular activities.

The authors document their findings through a combination of structural biology, biophysics and biochemical approaches. Particularly compelling to me are crystal structures showing co-occupancy of PPARg co-bound to the agonist edaglitazone and a nonanoic (C9) fatty acid at adjacent sites, with a concomitant ordering of the omega loop/AF2 surface involved in coactivator recruitment (Figure 3). Coupled with a retrospective analysis of other published PPARg/agonist structures identifying previously-nonmodeled fatty acid densities (Figure 5) and MD simulations showing linkage between ligand binding and omega loop rigidification (Figure 8), I find the authors' assertion of generality well-supported. Additional cell-based analyses of the effects of PPARg point mutations on transcriptional activity support the critical functional linkage between fatty acid binding and function (Figure 10).

Slightly less compelling are NMR data provided to support co-binding; while these are reasonable, they are also complicated by fairly subtle peak shifts and issues with presentation, the latter of which can be straightforwardly addressed in revision (Figures6, 7). The authors also valiantly work to provide in vitro biochemical data demonstrating the functional effects of co-binding despite the non-ideal situation posed by coupled binding of avid agonists and weakly-binding fatty acids (Figure 8), using covalently-bound antagonists to circumvent these challenges.

In summary, this work is significant by addressing an important issue in designing artificial ligands for this important receptor class, particularly in the proper design of in vitro or cellular conditions for functional studies. The data presented are clear for the most part (although could be improved in presentation or clarity, see comments below) and support the authors' assertions. I recommend publication after addressing these issues:

Figures 6, 7: The combination of subtle peak shifts and color selection (i.e. dark vs. light blue, or red, or green) in many of these panels often complicates the ability to see what shifts are or are not occurring upon ligand addition. Please consider alternative color choices and/or showing 1D projections, etc. to simplify.

Figure 8A: I agree with the authors that this is a critical biochemical experiment to validate the functional importance of coincident C9/agonist binding, but a couple of details are unclear. Was the protein delipidated beforehand? The EC50(C9) required for binding either TRAP220 or NCoR appears to be slightly and unexpectedly lower for the apo- protein vs. with either Rosi or Eda present (yellow vs. orange or blue curves), contrary to what would be expected of Rosi or Eda as PPARg agonists. Please clarify.

Figure 8—figure supplement 2: Please clarify why TR-FRET values are variable as a function of covalent antagonist binding, particularly i) very low TR-FRET values for T007… at low C9 values (in contrast to the other three samples) and ii) minimal changes in TR-FRET values are observed for SR168….

*Reviewer #3:*

This manuscript describes in great detail the alternative binding of natural and synthetic ligands to a non-orthesteric pocket at the surface of ppARgamma ligand binding domain. The authors confirm the presence of similar moieties in similar already published structures and continue to provide functional and in depth structural evidence (NMR and simulations) of an important regulatory role of these ligands in ppAR γ functioning. This is an excellent paper with eye for detail resulting in the in depth validation of a hypothesis which has been going around without much data to support it. This manuscript changes this.

I have no major comments. I am not a structural biologist, but the data and experiments have been described in sufficient detail and comprehensive way.

---

## [Author Response]

[Editors’ note: the author responses to the first round of peer review follow.]

How do synthetic ligands regulate the structure and function of nuclear receptors? This question has been the focus of many influential research efforts, including crystallography studies that have detailed the structure of the nuclear receptor ligand-binding domain bound to many different types of natural ligands and synthetic ligands. These structural studies have revealed that natural and synthetic ligands compete for binding to the same orthosteric pocket (i.e., one-for-one orthosteric ligand exchange), whereby a synthetic orthosteric ligand displaces a natural ligand from the orthosteric pocket. However, recent work by others and us have revealed that ligand binding to nuclear receptors can be more complex. One example includes the identification of an allosteric ligand-binding site in RORγ and the development of allosteric inhibitors that cobind with a natural ligand present in the orthosteric pocket (Nature Communications 2015, 6, 8833). In another example, we showed that synthetic ligands optimized to bind with high affinity to the PPARγ orthosteric pocket (site where natural ligands bind) are also able to bind to a second “alternate” ligand-binding site with lower affinity – even in the presence of an oxidized natural/endogenous lipid covalently bound to the orthosteric pocket (Nature Communications 2014, 5, 3571). This PPARγ alternate site for synthetic ligands has since been structurally and functionally validated by us and others in subsequent studies (cited in our manuscript). This current manuscript supports a new way of thinking about the mechanisms by which synthetic orthosteric ligands (including FDA-approved drugs) bind to PPARγ. In contrast to the widely accepted one-for-one orthosteric ligand exchange model, we show that synthetic orthosteric PPARγ ligands can cobind with, rather than completely displace, a natural/endogenous ligand. In a series of crystal structures (8 total; including 5 added in this revision), we provide mechanistic evidence that the orthosteric binding of the synthetic ligand can push natural/endogenous fatty acids out of the orthosteric pocket into the alternate site. Our NMR and MD data shows that cobinding of a synthetic ligand (to the orthosteric pocket) and natural/endogenous ligand (to the alternate site) has a cooperative/synergistic effect on stabilizing the coactivator binding surface in PPARγ. We also include new quantitative biochemical data (in response to the reviewer’s concerns) showing that ligand cobinding synergistically enhances coactivator interaction better than each ligand on its own. Furthermore, using mutagenesis and cellular assays, we show that synthetic and natural/endogenous ligand cobinding may affect PPARγ transcription in cells – we are careful to emphasize this as a possibility, and a limitation of our studies, because it is not possible to precisely define the natural ligand-bound state in cells as can be done in structural and biochemical studies. Our findings are different that the complex nuclear receptor ligand binding scenarios that we and others discovered previously. Our findings should influence the nuclear receptor field and researchers in other fields (e.g., GPCRs, kinases, etc.) to think outside the box – or, outside the orthosteric pocket – in understanding how synthetic orthosteric ligands may cobind synergistically with natural ligands to elicit pharmacological functions. In support, we would like to emphasize a point raised by reviewer 3 on our initial *eLife* submission, who wrote: "This is an excellent paper with eye for detail resulting in the in depth validation of a hypothesis which has been going around without much data to support it. This manuscript changes this." We believe this comment illustrates why our work is well suited for *eLife*.

While recognizing the quality of the work and its thoroughness, the principal concern regarding further review at eLife are issues of novelty and the level of new conceptual advance, particularly given the depth of information already available for this class of receptors.

We hope that our revised manuscript and the new data collected in response to the previous review addresses this principal concern. One thing we would like to point out first is that our previous manuscript focused on one medium chain fatty acid (MCFA), nonanoic acid (C9), which happened to copurify with our protein that we expressed and purified from bacteria. We performed additional studies with added (exogenous) C9, but in the previous review reviewer 1 raised the question on whether C9 is a relevant mammalian PPARγ natural ligand. In our revised manuscript, we point out that MCFAs are not only present in bacteria; they are also dietary mammalian natural PPARγ ligands obtained from foods such as oils and dairy products that display high serum concentrations (µM–mM) and activate PPARγ transcription, regulate the expression of PPARγ target genes, and influence cellular differentiation, adipogenesis, and insulin sensitization in mice.

Furthermore, we have expanded the scope of our manuscript by studying cobinding with two long-chain unsaturated fatty acids (UFAs), arachidonic acid (AA) and oleic acid (OA), which were shown using metabolomics studies to be endogenous ligands present in 3T3-L1 preadipocyte cells (a cell line relevant to PPARγ functions in fat tissue) and mouse brain and adipose tissue that bind to PPARγ (Kim, Lou and Saghatelian, 2011). In a series of 5 new crystal structures that we added to this revision (now there are 8 total), we provide mechanistic evidence that the orthosteric binding of a synthetic TZD ligand can push an UFA out of the orthosteric pocket into the alternate site. We did this by first solving co-crystal structures of UFA-bound PPARγ LBD, which showed the UFA bound to the orthosteric pocket; then by soaking in a synthetic ligand (rosiglitazone or GW9662) into the preformed UFA-bound crystals, which showed that the synthetic ligand cobinds with the UFA by moving it to an alternate site within the crystals (i.e., not the orthosteric binding mode obtained when the UFA was co-crystallized alone).

The idea that a natural ligand can synergize with a synthetic ligand to activate the receptor is not rigorously supported by the data in this paper.

We include new quantitative biochemical data that we believe addresses this concern. Using TR-FRET ligand-dependent recruitment assays coupled with mutagenesis (Figure 10) and fluorescence polarization (FP) binding assays (Figure 10—figure supplement 2), we show that natural ligand (C9, AA, and OA) cobinding with covalent orthosteric synthetic ligands (and therefore cannot be displaced) increases the interaction between the PPARγ ligand-binding domain (LBD) and a peptide derived from the TRAP220 coactivator protein. We also included TR-FRET data for two other MCFAs, C10 and C12 (Figure 10—figure supplement 1). Furthermore, we include new isothermal titration calorimetry (ITC) data to show that natural ligand (C9 or OA) cobinding with two different noncovalent synthetic orthosteric ligands (rosiglitazone or edaglitazone) has a synergistic effect on strengthening the binding affinity of the TRAP220 peptide through a favorable change in entropy (ΔS). Combined these data show that the nature and synthetic ligand cobinding scenarios that we captured in our crystal structures are functionally relevant and enhanced coactivator interaction.

A number of structural studies have already convincingly demonstrated the ability of nuclear receptors to be occupied by multiple ligand species (the authors appropriately cited several of these). Moreover, this is not the first structure to report that nonanoic acid can be co-bound to PPARg with other similar ligands (PLoS One 7:e36297, 2012). Prior papers (Hughes et al., 2014 and Puhl et al., 2012 discussed in the third paragraph of the Discussion) detail a mix of natural and artificial compounds binding concurrently to PPARg's orthosteric pocket and the alternative site, presenting structural and cell biology data to support the nature of the interaction and functional relevance.

Two issues were raised here: (i) our findings have a potential lack of novelty relative to other studies that have shown multiple ligands can bind to nuclear receptors; and (ii) our manuscript lacked cell biology functional data support that ligand cobinding has functional relevance.

In addressing (i), we believe our findings do have novelty, and we revised our discussion to describe how our findings are related to previous studies but also different and novel. Related to specific studies cited in the editorial summary:

Our previous paper (Hughes et al.) showed that synthetic ligands designed to bind to the orthosteric pocket can also bind to a second alternate site. Our work in this current manuscript is different – essentially the opposite scenario – and our findings here are even more interesting because it challenges the way the field thinks about ligand binding to nuclear receptors (i.e., does it occur via orthosteric ligand exchange with a natural ligand, or via cobinding with a natural ligand whereby the natural ligand is pushed out of the orthosteric pocket into an alternate site).

The other paper cited (Puhl et al.) showed a crystal structure of the luteolin (a natural product isolated from plants) bound to the alternate site with a cobound bacterial fatty acid (myristic acid) in the orthosteric pocket. The Puhl et al. study is more related to our previous paper (Hughes et al.), and there is an important difference of both studies with our current work. The current thinking in the field is that the synthetic TZDs/drugs we studied in this current work (and arguably all other synthetically designed PPARγ ligands) completely displace (i.e., cause unbinding of) natural ligands from the orthosteric pocket when the synthetic ligand binds to the orthosteric pocket. Our work here supports a new way of thinking about the mechanisms by which synthetic orthosteric ligands (including FDA-approved drugs) bind to PPARγ. As reviewer 3 stated (see their comments below), we hope that our work will make the field think differently about the mechanism of ligand/drug binding to PPARγ and other drug targets. – We believe this part of the editorial summary is not correct: "this is not the first structure to report that nonanoic acid can be co-bound to PPARg with other similar ligands (PLoS One 7:e36297, 2012)". The referenced PLoS One paper does not show cobinding binding natural and synthetic ligands, but instead reports and compares two crystal structures: (1) a structure with three nonanoic acid molecules bound to the orthosteric pocket, and (2) a structure of rosiglitazone bound to the orthosteric pocket. Furthermore, we searched the PDB and primary literature and could not find any examples of published studies with findings similar to the results we present in this manuscript.

In addressing (ii), we acknowledge that one limitation of our work is that it is difficult to extend our quantitative structural and biochemical findings on how endogenous ligands that are present in cells may cooperatively affect PPARγ function using cellular studies. In biochemical and structural studies, the amount of synthetic and natural/endogenous ligands can precisely controlled; this is simply not possible to do in cells because there is no way to precisely control all endogenous ligands present within cells. In contrast, it is possible to control the amount of synthetic (non-natural) ligands in cells, which makes it easier to validate orthosteric and alternate site of the same ligand or different synthetic ligands in cells, as we did in our previous paper (Hughes et al.).

Notwithstanding these issues, in response to this concern, we added some new cell-based transcription data showing that addition of exogenous UFAs, which are also present in cells, does not impact PPARγ transcription. We therefore interpreted the cell-based transcription data (Figure 11) with the premise that PPARγ will be bound to and activated by endogenous ligands present within the cells. We revised the Results section describing these cellular transcription data to acknowledge the limitations and also to better inform the readers (and reviewers) on how we devised the experiments and interpreted the data. We are also careful to state that synthetic and natural/endogenous ligand cobinding *may* affect PPARγ transcription in cells – we emphasize this as a possibility, and a limitation of our studies, because as mentioned it is not possible to precisely define the natural ligand-bound state in cells as can be done in structural and biochemical studies.

Finally, beyond PPARg, it is not clear how broadly this concept might apply.

We revised our Discussion extensively to detail how our findings could have a broader impact to the nuclear receptor field and beyond. We include references to studies on 15 different nuclear receptors that have been shown to bind to fatty acids, many of which are not recognized as lipid-binding receptors (e.g., steroid receptors such as AR, ER, FXR, GR, LXR, PR, and TR), as well as studies showing fatty acid interactions with kinases, GPCRs, and ion channels. Furthermore, we cite metabolomics studies suggesting that most proteins likely bind endogenous metabolites in cells.

Specific points raised by the reviewers are appended below, which may be helpful in revising the manuscript for submission elsewhere.

Reviewer 1:

1) The structure reveals the bacterial lipid, nonanoic acid, bound to PPARg, a finding that has been shown by others as the authors note. However, this finding appears to be an artifact of preparing the protein in bacteria, an artifact that has been observed in numerous other nuclear receptor structures. Importantly, however, no evidence is given to support the idea that nonanoic acid is a naturally occurring ligand at relevant concentrations in mammalian cells. The authors also show one set of data using C10 and C12 lipids that can bind competitively to the pocket, but no data were presented to show they function similarly to the C9 lipid.

We addressed most of this concern in our first response paragraph above, but in summary we cite published evidence that MCFAs are dietary ligands present at relevant concentrations in human serum and activate PPARγ transcription. In addition to the data on other MCFAs (C10 and C12), we include new data on the UFAs AA and OA, which are endogenous PPARγ ligands present in 3T3-L1 preadipocyte cells (a cell line relevant to PPARγ functions in fat tissue) and mouse brain and adipose tissue that bind to PPARγ (Kim, Lou and Saghatelian, 2011). Our studies with UFAs include 5 new crystal structures, NMR data, TR-FRET data, FP data, and ITC, which combined show that UFA cobinding with a synthetic TZD affects the structure and function of PPARγ.

2) The data in Figure 10 might support the idea of endogenous cellular lipids bound to PPARg, but again this was not proven and there could be other interpretations of these results. These mutations may simply compromise basal activity due to non-ligand dependent changes in the structure of the pocket, similar to that revealed in other nuclear receptors.

Here the concern is that the mutant variants of PPARγ that we transfected into cells, which showed lower transcriptional activities than wild-type PPARγ, may be due effects on function other than affecting natural ligand binding. (Please note that the previous Figure 10 is now Figure 11 in our revised manuscript). It is true that mutations can have multiple effects on receptor function. We showed that lower activities of the mutants is not due to lower protein expression levels in cells via western blot (Figure 11—figure supplement 1) or coregulator binding affinity (Figure 11—figure supplement 2). Furthermore, the mutants did not affect the binding affinity or relative activation window of synthetic ligands edaglitazone or rosligtazone (all mutants have similar EC50 values vs. wild-type). However, our TR-FRET data show that the mutants do reduce natural ligand binding (Figure 10) in particular when the orthosteric pocket is “blocked” (i.e., when PPARγ is bound to a covalent orthosteric ligand). Taken together, this indicates that the mutants likely inhibit the alternate site natural ligand binding mode, suggesting that the lower activities in cells may be due to inhibited natural ligand binding in cells. This concept is supported by our quantitative structural and biochemical studies. However, given that it is not possible to precisely define the natural ligand-bound state in cells as can be done in structural and biochemical studies, we were careful to include the more careful statement that the data suggest that the cobinding of endogenous ligands and synthetic TZDs *may* synergistically activate PPARγ transcription over the activity of a single ligand alone. We also revised the section describing the cellular data in relationship to our quantitative biochemical and structural studies, including the mutant variants we studied, to better inform on how we devised the experiments and interpreted the data.

Reviewer 2:

Figures 6, 7: The combination of subtle peak shifts and color selection (i.e. dark vs. light blue, or red, or green) in many of these panels often complicates the ability to see what shifts are or are not occurring upon ligand addition. Please consider alternative color choices and/or showing 1D projections, etc. to simplify.

We thank the reviewer for pointing this out and agree. We have changed the color scheme to hopefully better differentiate between the three NMR peaks.

Figure 8A: I agree with the authors that this is a critical biochemical experiment to validate the functional importance of coincident C9/agonist binding, but a couple of details are unclear. Was the protein delipidated beforehand? The EC50(C9) required for binding either TRAP220 or NCoR appears to be slightly and unexpectedly lower for the apo- protein vs. with either Rosi or Eda present (yellow vs. orange or blue curves), contrary to what would be expected of Rosi or Eda as PPARg agonists. Please clarify.

In this experiment shown in the previous version of our manuscript, we attempted to use the TR-FRET assay to show cobinding of C9 and a non-covalent synthetic ligand. In retrospect, this was not the best experimental approach because it required saturating concentrations of the synthetic ligand, which effectively maxed out the TR-FRET window of response. We removed these data from the revised manuscript and replaced it with ITC data. As mentioned in the Results section describing these data, the nature of ITC experiment allows nearly complete control of the experimental setup, in particular the ability to precisely define the components of the assay (e.g., stoichiometry of UFA and TZD ligands in the sample cell). Our ITC results (Table 1) show that ligand cobinding synergistically enhances coactivator interaction better than each ligand on its own. The protein we used for the TR-FRET assays, which requires very low concentrations of protein (~4 nM), was delipidated using the LIPIDEX protocol described in our methods. However, protein used for FP and ITC experiments was not delipidated using the LIPIDEX protocol because the protocol results in a significant loss (~50% or more) of protein; instead the protein was subjected to an extensive column chromatography purification procedure that significantly reduces the amount of (putatively) bound bacterial lipids. This was verified by NMR as well as ITC and FP experiments where the addition of C9 (or any other fatty acid; e.g., AA or OA) to the protein caused a strengthening of the coactivator binding affinity vs. apo-protein as detected by ITC (Table 1); no change in coactivator affinity would be expected if the protein was bound/saturated by bacterial lipid. We have included this information in our revised Materials and methods section under protein purification.

Figure 8—figure supplement 2: Please clarify why TR-FRET values are variable as a function of covalent antagonist binding, particularly i) very low TR-FRET values for T007… at low C9 values (in contrast to the other three samples) and ii) minimal changes in TR-FRET values are observed for SR168….

In reference to (i), the initial TR-FRET values were very low for the T0070907 (T007) ligand because, as we recently discovered in a manuscript accepted at Nature Communications (available at https://www.biorxiv.org/content/early/2018/01/17/245852 and https://www.nature.com/articles/s41467-018-07133-w), T007 is not an “antagonist”. By definition, an antagonist should not change coregulator affinity to a large degree. Instead, our work showed that T007 is a corepressor-selective inverse agonist that represses PPARγ transcription. We removed the T007 data from this manuscript since it has more complicated pharmacology and we simply wanted to use the covalent ligands to block the orthosteric pocket.

In reference to (ii), there are minimal changes in TR-FRET for the C9, AA, and OA dose responses PPARγ is covalently bound to SR16832 because it contains an extension towards the alternate site that weakens/inhibits alternate site ligand binding (Brust et al., 2017). This is shown visually in our crystal structures shown in Figure 8B, C.

We also included this paragraph in the results describing the TR-FRET data:

“In the TR-FRET assay, the overall window of activity is related to the change in binding affinity of the coactivator peptide; a larger increase in the assay window indicates a larger change in affinity. Because coregulator binding affinity is influenced differently by covalent ligands (Brust et al., 2017), which causes differences in the basal TR-FRET response (i.e., at low ligand concentrations or DMSO control), we normalized the TR-FRET ratio to easily compare the ligand cobinding and the change in the assay window among the conditions tested.”

Subsection “Luciferase reporter assay and western analysis of protein levels”: replace "luciverase" with "luciferase".

Thank you for pointing this out, we made this change.

Figure 4C: consider showing location of Phe363 and Met364 here to facilitate links with Figure 4D.

Thank you for pointing this out, we made this change (now Figure 3D in this revision).

Figure 6—figure supplements 3-5: change "Related to Figure 5A, 5B, 5C…" to "6A, 6B, 6C…"? Figure 6—figure supplement 8: what do the schematic dotted line peaks represent? Not described.

Thank you for pointing this out, we made sure that the “Related to” callouts in the supplementary figures match the correct main figures. We also added a note as to what the schematic dotted peaks represent (Figure 5—figure supplement 5 in this revision): “The black arrow denotes a peak present when PPARγ is bound to C9; grey arrows denote the lack of transition to a C9-bound peak, which is missing in the rosiglitazone-bound data (dotted circle/cross).”

Reviewer 3:

[…] There is no reference to the Chandra et al., 2017 paper in which a co-crystal of ppARgamma-RXRalpha bound to DNA was solved. I suggest to include in the Discussion whether or not similar lipids can be detected in this structure and whether this would/could affect the communication between LBD and DBDs.

The 3 crystal structures reported in the Chandra et al., 2017 study (PDB: 3dzu, 3dzy, 3e00) were 3 out of 29 structures we examined for putative density consistent with a cobound bacterial MCFA. We did not see any density in the three crystal structures, which may be due to the relatively lower resolution of these structures (3.2Å, 3.1Å, and 3.1Å, respectively) compared to the relatively high resolution for the other structures were we observed the density shown in Figure 4 (2.1–2.2Å). In the Discussion of our revised manuscript, we added the following statement about the potential for allosteric communication as suggested by the reviewer: “The cobound fatty acid in the alternate site, which is located near the β-strand surface and Ω-loop region, is structurally close to two other important surfaces. In a crystal structure of the intact PPARγ/RXRα heterodimer bound to DNA (47), there is a long-range interaction between this region in PPARγ and the RXRα DNAbinding domain (DBD). It is possible that ligand cobinding could allosterically affect or synergize with this long-range interaction.”